# From Associations to Activations: Comparing Behavioral and Hidden-State Semantic Geometry in LLMs

**Louis Schiekiera** [1][2]  **Max Zimmer** [3][4]  **Christophe Roux** [3][4]  **Sebastian Pokutta** [3][4]  **Fritz Günther** [1]

## Abstract

We investigate the extent to which an LLM's hidden-state geometry can be recovered from its behavior in psycholinguistic experiments. Across eight instruction-tuned transformer models, we run two experimental paradigms—similarity-based forced choice and free association—over a shared 5,000-word vocabulary, collecting 17.5M+ trials to build behavior-based similarity matrices. Using representational similarity analysis, we compare behavioral geometries to layerwise hidden-state similarity and benchmark against FastText, BERT, and cross-model consensus. We find that forced-choice behavior aligns substantially more with hidden-state geometry than free association. In a held-out-words regression, behavioral similarity (especially forced choice) predicts unseen hidden-state similarities beyond lexical baselines and cross-model consensus, indicating that behavior-only measurements retain recoverable information about internal semantic geometry. Finally, we discuss implications for the ability of behavioral tasks to uncover hidden cognitive states.

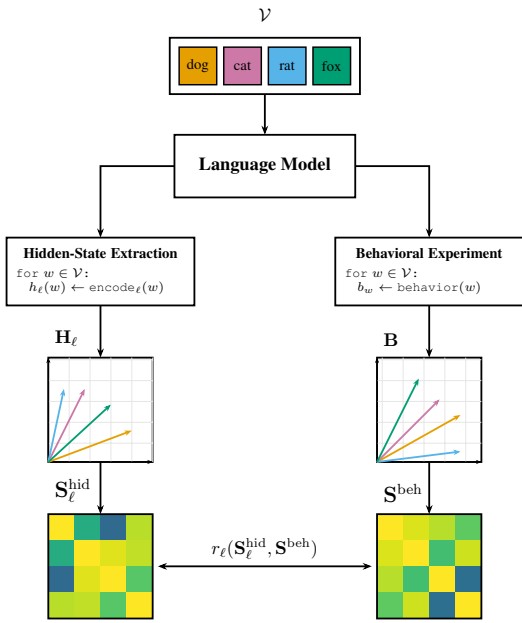

*Figure 1.* Conceptual overview. For a shared vocabulary $\mathcal{V}$, we (i) extract layer-$\ell$ word representations to form a hidden-state similarity matrix $\mathbf{S}_\ell^{\mathrm{hid}}$, and (ii) run behavioral association tasks (forced choice/free association) to build a cue–response matrix $\mathbf{B}$ and behavioral similarity $\mathbf{S}^{\mathrm{beh}}$. RSA correlates the pairwise similarities in $\mathbf{S}_\ell^{\mathrm{hid}}$ and $\mathbf{S}^{\mathrm{beh}}$ to quantify behavior–activation alignment.

## 1. Introduction

In cognitive science, semantic knowledge is typically treated as a latent structure: we cannot observe a speaker's 'meaning representation' directly, but we can systematically probe it through behavior (De Deyne et al., 2019; Günther et al., 2019; Jones et al., 2015). Word-association paradigms use this measurement logic: when a participant sees a cue (e.g.

dog), the associations they produce or select (e.g. *cat*, *leash*, *bark*) are constrained by their underlying semantic organization. When such judgments are aggregated across trials, the resulting cue–response statistics are used for inference: cues that show similar response distributions are inferred to be semantically close, yielding an embedding-like similarity matrix, often conceptualized as a structured mental lexicon or semantic network (De Deyne & Storms, 2008; De Deyne et al., 2013; Roads & Love, 2021; Vankrunkelsven et al., 2018). In this sense, association behavior functions as a measurement device: it produces observable data from which one can reconstruct an approximate map of an otherwise unobserved semantic system.

We transfer this measurement logic to large language models (LLMs). Recent work increasingly treats LLMs as 'participants' in classic semantic paradigms, using free association

[1]Computational Modelling Lab, Humboldt-Universität zu Berlin, Berlin, Germany [2]Department of Education and Psychology, Freie Universität Berlin, Berlin, Germany [3]Department for AI in Society, Science, and Technology, Zuse Institute Berlin, Berlin, Germany [4]Institute of Mathematics, Technische Universität Berlin, Berlin, Germany. Correspondence to: Louis Schiekiera <louis.schiekiera@hu-berlin.de>.

*Proceedings of the 43rd International Conference on Machine Learning*, Seoul, South Korea. PMLR 306, 2026. Copyright 2026 by the author(s).

and related protocols to construct model-derived semantic norms and network structure that can be compared to large-scale human datasets (Abramski et al., 2024; 2025; Suresh et al., 2023; Vintar & Javoršek, 2025). A key open question, however, is not only how model behavior compares to humans, but also what a model's *own* behavior reveals about its *own* internal representations.

This question is now empirically testable because, unlike in humans, both behavior *and* internal representations are observable in LLMs (Jawahar et al., 2019; Tenney et al., 2019; Zhang et al., 2023). Figure 1 summarizes our approach: we probe a model over a shared vocabulary, derive a behavioral semantic geometry from its responses, and then compare that geometry to the model's layerwise hidden-state geometry. Concretely, by repeatedly querying a model with a controlled vocabulary and aggregating responses across many trials, we obtain for each cue $w_i$ a response distribution encoded as a row $\mathbf{B}_{i,:}$ of a cue–response matrix $\mathbf{B}$. Each row thus defines a behavioral embedding, and comparing rows induces a behavioral similarity geometry, e.g., $\mathbf{S}^{\text{beh}}(i,j) = \cos(\mathbf{B}_{i,:}, \mathbf{B}_{j,:})$. Our analysis then asks how well $\mathbf{S}^{\text{beh}}$ recovers the hidden-state similarities $\mathbf{S}^{\text{hid}}_{\ell}$ across layers and prompting contexts. This comparison is useful in practical settings where only black-box behavioral access is available, because it tests how much of a model's internal semantic organization is recoverable from discrete outputs.

Representational Similarity Analysis (RSA) provides a solution to compare representations that differ in dimensionality, scaling, and modality (e.g. behavior, and neural data): rather than aligning coordinates, RSA compares the *geometry* of two representational spaces by correlating their pairwise similarity structure over a shared set of items (Kriegeskorte et al., 2008; Nili et al., 2014). RSA has been widely used to relate representations across modalities (Braun et al., 2025; Ciernik et al., 2025; Klabunde et al., 2025; Kornblith et al., 2019; Sucholutsky et al., 2024), including comparisons between LLM activations and human brain signals (Abnar et al., 2019; Aw et al., 2024). However, to the best of our knowledge, RSA has not been used to directly compare an LLM's *behavior-derived* semantic geometry with its *own* layerwise hidden-state geometry under a matched vocabulary and experimental protocol.

In this work, we propose a framework to compare an LLM's behavioral semantic geometry with its internal hidden-state geometry. Across eight instruction-tuned transformer models, we use two psycholinguistic paradigms—free association (FA) and forced choice (FC)—to collect semantic relations from model behavior and construct behavioral embedding matrices. In parallel, we extract hidden-state representations for the same vocabulary across layers and multiple extraction strategies. This paired design enables within-model alignment between behavior and internals.

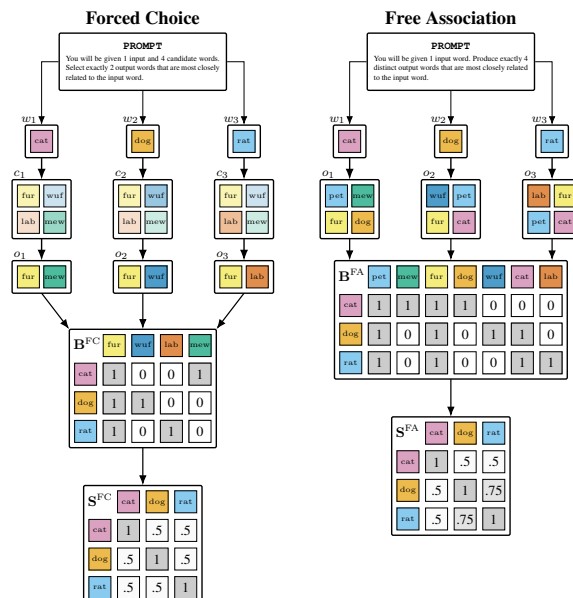

*Figure 2.* Behavioral paradigms and derived semantic geometries. Left (forced choice): given a cue word $w_i$ and a candidate set $c_i$, the model selects a fixed number of output words $o_i$, producing a cue–response count matrix $\mathbf{B}^{\text{FC}}$. Right (free association): given $w_i$ alone, the model generates multiple output words $o_i$, yielding $\mathbf{B}^{\text{FA}}$. From the count matrix, we produce similarity matrices $\mathbf{S}^{\text{FC}}$ and $\mathbf{S}^{\text{FA}}$ by cosine similarity between rows. The diagram shows $|c_i| = 4$ for FC and $|o_i| = 4$ for FA, while our experiments use $|c_i| = 16$ for FC and $|o_i| = 5$ for FA.

We evaluate alignment using RSA and a complementary encoding analysis, asking at which layers and under which prompting conditions an LLM's internal representations most closely reflect the semantics it expresses behaviorally.

Our contributions are:

1. **Behavior–Activation Alignment.** We compare behavior-derived semantic geometries from FC and FA to layerwise hidden-state geometry across eight instruction-tuned transformer models using RSA and nearest-neighbor overlap. We provide a prompt- and layer-resolved characterization of when internal similarity structure matches behavioral semantics.

2. **Predictability from Behavior.** Using a held-out-words ridge regression protocol, we show that behavioral similarity from discrete outputs—especially FC—predicts unseen hidden-state similarities beyond lexical baselines (FastText, BERT) and a cross-model consensus reference.

3. **Task Design and Matrix Density.** Forced choice substantially outperforms free association across all measures. We attribute this gap to the structure of the elicited data: FC restricts each cue's responses

to a shared, fixed set of candidate words, yielding a dense cue–response matrix $\mathbf{B}^{\mathrm{FC}}$, whereas FA lets responses range over an open vocabulary, producing a wide, sparse, heavy-tailed $\mathbf{B}^{\mathrm{FA}}$ with little overlap between cues.

4. **Dataset.** We release a large-scale dataset of LLM association behavior (17.5M+ trials) spanning FA and FC across eight instruction-tuned models, intended as a reusable resource for behavioral interpretability and representational-alignment research.

## 2. Related Work

A growing line of work uses LLMs to generate semantic norms and association networks that can be compared to large-scale human resources such as *Small World of Words* (Abramski et al., 2024; 2025; Suresh et al., 2023; Vintar & Javoršek, 2025). These studies show that task-elicited semantic structure from LLM outputs often exhibits meaningful overlap with human judgments, while also revealing systematic divergences that reflect model-specific biases (Abramski et al., 2024; Suresh et al., 2023).

Prior analyses of transformer representations show that linguistic and semantic information is accessible from hidden states and varies systematically across depth (Derby et al., 2021; Liu et al., 2024; Lenci et al., 2022; Tenney et al., 2019). Other work relates model activations to external measurements, including brain activity and behavioral signals (Abnar et al., 2019; Aw et al., 2024). Our contribution differs in focusing on alignment between (i) behavioral semantic geometry and (ii) layerwise hidden-state similarity.

Work on extraction and cloning attacks reconstructs internal components of LLMs from API outputs, typically assuming access to logits or log-probabilities (Carlini et al., 2024; Gharami et al., 2025). Our setting is deliberately weaker: we use discrete association judgments (no logits) to ask what aspects of internal *similarity geometry* are recoverable. Finally, evidence for shared structure across LLMs motivates a low-dimensional 'universal' or 'platonic' semantic geometry (Huh et al., 2024; Jha et al., 2025; Kaushik et al., 2025); we capture this with a cross-model consensus baseline to separate shared from behavior-specific structure.

## 3. Methods

### 3.1. Vocabulary and Preprocessing

We begin from the SUBTLEX-US lexicon (Brysbaert et al., 2012) and construct a core noun vocabulary by part-of-speech filtering, lemmatization, and lemma deduplication, then select the top 6,000 nouns by frequency. We then intersect this list with the C4 corpus by retrieving 50 sentences per word; the final vocabulary consists of the 5,000 highest-

*Table 1.* Model specifications. $n_{\mathrm{params}}$ = number of parameters in billions (B); $n_{\mathrm{layers}}$ = number of layers; $d_{\mathrm{model}}$ = hidden-state dimension width. HuggingFace Model IDs are reported in Appendix B.

| Model | $n_{\mathrm{params}}$ | $n_{\mathrm{layers}}$ | $d_{\mathrm{model}}$ |
|---|---|---|---|
| Falcon3-10B-Instruct | 10B | 40 | 3072 |
| gemma-2-9b-it | 9B | 42 | 3584 |
| Llama-3.1-8B-Instruct | 8B | 32 | 4096 |
| Mistral-7B-Instruct-v0.2 | 7B | 32 | 4096 |
| Mistral-Nemo-Instruct-2407 | 12B | 40 | 5120 |
| phi-4 | 14B | 40 | 5120 |
| Qwen2.5-7B-Instruct | 7B | 28 | 3584 |
| rnj-1-instruct | 8B | 32 | 4096 |

*Table 2.* Data collection statistics for the two behavioral paradigms across eight models and a shared vocabulary of 5,000 words. $\mathrm{T_{total}}$ = total number of trials, $\mathrm{T_m}$ = per model, $\mathrm{T_w}$ = per input word, and $\mathrm{T_{w+m}}$ = per model and word.

| Paradigm | $\mathrm{T_{total}}$ | $\mathrm{T_m}$ | $\mathrm{T_w}$ | $\mathrm{T_{w+m}}$ |
|---|---|---|---|---|
| Forced choice | 12.52M | 1.565M | 2504 | 313 |
| Free association | 5.04M | 0.630M | 1008 | 126 |

frequency nouns for which 50 C4 sentences are available (Raffel et al., 2020; Tikhomirova & Wulff, 2026). Further details on preprocessing are provided in Appendix C.

### 3.2. Models

We evaluate eight instruction-tuned decoder-only transformer models (see Table 1). The models include Falcon3-10B-Instruct (TII Team, 2024), gemma-2-9b-it (Gemma Team, 2024), Llama-3.1-8B-Instruct (Meta AI, 2024), Mistral-7B-Instruct-v0.2 (Jiang et al., 2023), Mistral-Nemo-Instruct-2407 (Mistral AI, 2024), phi-4 (Abdin et al., 2024), Qwen2.5-7B-Instruct (Qwen Team, 2024), and rnj-1-instruct (Vaswani et al., 2025).

### 3.3. Behavioral Association Paradigms

Figure 2 summarizes the two behavioral paradigms used to produce semantic association structure from each model. Table 2 provides statistics on the number of trials collected for each paradigm. Both paradigms operate over the same fixed vocabulary of 5,000 nouns.

#### 3.3.1. FORCED-CHOICE PARADIGM

Forced-choice tasks are a standard tool in cognitive psychology and psycholinguistics for studying semantic similarity under fixed candidate sets (Demiralp et al., 2014; Günther et al., 2023; Li et al., 2016; Roads & Love, 2021; Tversky, 1977). Compared to free-response tasks, forced choice restricts responses to a predefined set that can include both

weakly related and unrelated distractors, thereby probing relative similarity over a broad range of association strengths (De Deyne et al., 2012). In our FC paradigm, each cue word $w_i$ is presented together with 16 candidate words, from which the model must select exactly two words that are most semantically related to the cue (see Appendix D.1 for the full prompt). Candidate sets are constructed by a deterministic shuffle of the remaining 4,999 words using a cue-specific random seed (one seed per cue). This results in $\lceil \frac{4,999}{16} \rceil = 313$ FC trials per cue. Each trial produces a sparse binary signal indicating which two of the 16 candidates are most similar to the cue; aggregating across the 313 partitions ensures that every other vocabulary item co-occurs with the cue in exactly one trial, yielding the dense cue–cue structure.

### 3.3.2. FREE ASSOCIATION PARADIGM

In contrast to forced-choice tasks, free association places minimal constraints on responses, allowing participants to generate whatever associates come most readily to mind (De Deyne et al., 2019). As a result, FA norms capture aspects of semantic centrality, and have been widely used to study semantic networks, and spreading activation (Aeschbach et al., 2025; De Deyne et al., 2019; Petrenco & Günther, 2025). Recently, Abramski et al. (2024) collected a dataset of free associations of three LLMs. In the free association paradigm, the model is prompted with a single cue word $w_i$ and asked to generate exactly five single-word associates (see Appendix E.1 for the full prompt). To obtain a comparable number of associations per cue word as in the FC paradigm, we repeat this task across multiple stochastic runs with different random seeds. Specifically, we perform 126 runs per cue word.

### 3.3.3. POSTPROCESSING

Both paradigms were designed to yield similar association counts per cue (FC: 626; FA: 630). We excluded non-compliant outputs (e.g., out-of-set selections in FC, cue repetition in FC/FA); for FC, we issued a repair prompt and retried up to five times when needed. After postprocessing, mean usable associations per cue were 610.1 (97.5%) for FC and 622.6 (98.8%) for FA. Compliance details are in Appendix D.3 and Appendix E.3. All behavioral similarity matrices are computed from compliant associations only.

For each paradigm, we aggregate model outputs into a sparse cue–response count matrix $\mathbf{B}$, with rows indexing cue words and columns indexing response types. We write $\mathbf{B}^{FC}$ for the forced-choice matrix and $\mathbf{B}^{FA}$ for the free-association matrix. To reduce the influence of globally frequent responses, we reweight cue–response counts with positive pointwise mutual information (PPMI; see Appendix C.3), which emphasizes informative co-occurrences (Abramski et al., 2024).

Finally, we compute a cue–cue similarity matrix by taking cosine similarity between the PPMI-weighted row vectors.

### 3.4. Hidden-State Extraction Strategies

An important design decision in representational analyses of language models concerns the task context in which word-level hidden states are extracted (Bommasani et al., 2020; Cassani et al., 2024; Chronis & Erk, 2020; Gurnee & Tegmark, 2023; Tikhomirova & Wulff, 2026). Building on prior work, we extract layerwise word representations under four *contextual embedding strategies*. For each model and each target word $w_i$, we consider the following strategies:

- **Averaged.** The target word embedded in 50 naturally occurring sentences sampled from the C4 corpus (Raffel et al., 2020). Hidden states are extracted separately for each sentence and then averaged, resulting in a context-aggregated representation (Bommasani et al., 2020; Cassani et al., 2024; Tikhomirova & Wulff, 2026). For further details see Appendix C.

- **Meaning.** A single fixed, definition-style prompt (`'What is the meaning of the word {w}?'`), providing a minimal but explicit semantic context (Tikhomirova & Wulff, 2026).

- **Task (FC).** The target word embedded in the instruction prompt used for the forced-choice behavioral paradigm without the candidate list (see Appendix D.1 for the full prompt).

- **Task (FA).** The target word embedded in the instruction prompt used for the free-association paradigm (see Appendix E.1 for the full prompt).

Let $h_\ell(w, c)$ denote the residual-stream hidden state, i.e., the post-block representation returned after transformer block $\ell$ (self-attention + MLP), of word $w$ in context $c$ of a decoder-only transformer where $c$ specifies the full textual input provided to the model.

We define the extracted word representation at layer $\ell$ under strategy $s$ as $\mathbf{e}_\ell^s(w)$, computed as follows. For single-context strategies, $\mathbf{e}_\ell^s(w) = h_\ell(w, c_s(w))$, where $c_s(w)$ denotes the strategy-specific prompt in which $w$ appears. For the *Averaged* strategy, we follow prior work and aggregate across multiple natural contexts:

$$\mathbf{e}_\ell^{(\text{avg})}(w) = \frac{1}{50} \sum_{i=1}^{50} h_\ell(w, c_i(w)),$$

where each $c_i(w)$ is a distinct sentence sampled from the C4 corpus that contains $w$ (Bommasani et al., 2020; Tikhomirova & Wulff, 2026). For words split into multiple subword tokens, we average hidden states over the token

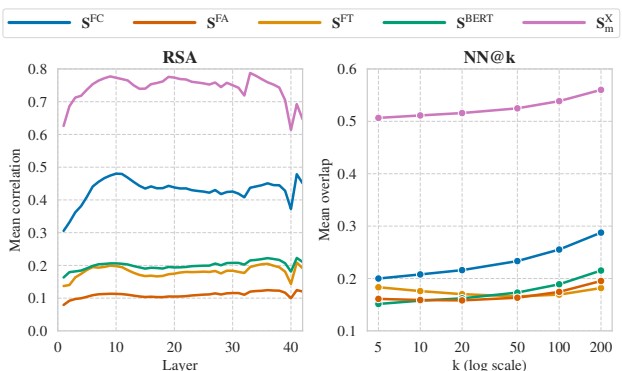 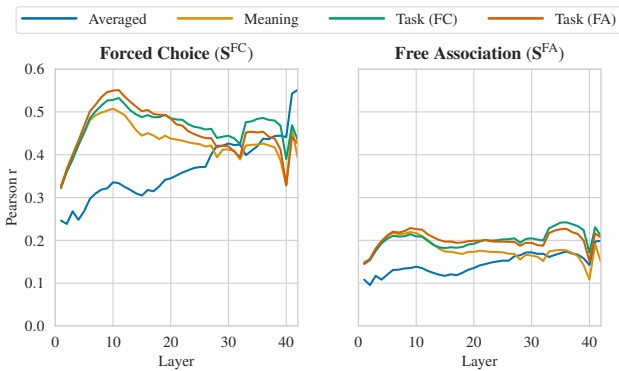

*Figure 3.* Summary of RSA and neighborhood-overlap results (means across models). *Panel a (left)* compares multiple reference geometries: *(a1)* mean RSA Pearson correlation as a function of layer, and *(a2)* mean nearest-neighbor overlap (NN@$k$) as a function of neighborhood size $k$ (log scale). *Panel b (right)* focuses on behavioral references and compares extraction strategies: *(b1)* layerwise RSA for PPMI-weighted forced-choice similarity $\mathbf{S}^{\text{FC}}$ and *(b2)* layerwise RSA for PPMI-weighted free-association similarity $\mathbf{S}^{\text{FA}}$.

positions whose offset spans overlap the cue's character span.

For each layer $\ell$, we then compute a hidden-state similarity matrix

$$\mathbf{S}_\ell^{\text{hid}}(i,j) = \cos\big(\mathbf{e}_\ell^s(w_i), \mathbf{e}_\ell^s(w_j)\big).$$

We exclude layer 0 because it consists of static, pre-transformer token embeddings that are not yet contextualized and therefore tend to reflect surface/lexical identity more than the contextual similarity structure we aim to analyze (Kumar et al., 2024; Cassani et al., 2024). In addition, contextual hidden-state spaces in transformers are known to be anisotropic: vectors concentrate in a narrow cone, so cosine similarity can be driven by shared global directions rather than item-specific semantic differences (Ethayarajh, 2019). To mitigate this, for each model, layer $\ell$, and extraction strategy $s$, we mean-center the extracted vectors by subtracting the empirical mean over the vocabulary before computing cosine similarity. Concretely, letting $\mathbf{e}_\ell^s(w_i) \in \mathbb{R}^d$ be the vector for word $w_i$ and $\mu_\ell^s = \frac{1}{|\mathcal{V}|}\sum_{i=1}^{|\mathcal{V}|} \mathbf{e}_\ell^s(w_i)$, we use $\widetilde{\mathbf{e}}_\ell^s(w_i) = \mathbf{e}_\ell^s(w_i) - \mu_\ell^s$ (Huang et al., 2021).

### 3.5. Baselines

Beyond behavioral embeddings, we compare hidden-state similarities to three vocabulary-aligned baselines (for further details see Appendix C.2).

- **FastText.** Pretrained English FastText vectors trained on Common Crawl (300d) (Bojanowski et al., 2017). We form a FastText similarity matrix $\mathbf{S}^{\text{FT}}$ by cosine similarity between the aligned word vectors.

- **BERT.** `bert-base-uncased`; we embed each word in a fixed base prompt and extract the mean of the

subword tokens aligned to the target word span from the final hidden layer (Devlin et al., 2019). We form a BERT similarity matrix $\mathbf{S}^{\text{BERT}}$ by cosine similarity between these word-level embeddings.

- **Cross-model consensus.** We define a cross-model consensus geometry by aggregating hidden-state cosine-similarity matrices across the remaining models (excluding the target model) to obtain a single reference similarity structure over the shared vocabulary. This baseline is motivated by recent evidence for a shared, low-dimensional semantic subspace across diverse LLMs, often discussed as a *universal* or *platonic* representational geometry (Huh et al., 2024; Jha et al., 2025; Kaushik et al., 2025; Wolfram & Schein, 2025; Lee et al., 2025). We define the cross-model consensus for target model $m$ as the mean pairwise cosine similarity across all layers of all *other* models:

$$s^{(m',\ell)}(i,j) := \cos\big(\mathbf{e}_{\ell,m'}^s(i), \mathbf{e}_{\ell,m'}^s(j)\big), \quad (1a)$$

$$\mathbf{S}_m^{\text{X}}(i,j) := \frac{1}{Z} \sum_{m' \neq m} \sum_\ell s^{(m',\ell)}(i,j). \quad (1b)$$

where $\mathbf{e}_{\ell,m'}^s(w)$ is the layer-$\ell$ word vector from model $m'$ under strategy $s$, and $Z$ is the total number of model–layer terms included. This reference excludes the target model to avoid leakage.

### 3.6. Evaluation

#### 3.6.1. REPRESENTATIONAL SIMILARITY ANALYSIS

RSA quantifies the extent to which different representational spaces share the same *pairwise similarity structure* (Kriegeskorte et al., 2008; Nili et al., 2014). For each model,

embedding extraction strategy, and transformer layer $\ell$, we compare the hidden-state similarity matrix $\mathbf{S}_\ell^{\text{hid}}$ to five reference semantic geometries defined over the same vocabulary: (i) PPMI-weighted forced-choice behavioral similarity $\mathbf{S}_{\text{PPMI}}^{\text{FC}}$, (ii) PPMI-weighted free-association behavioral similarity $\mathbf{S}_{\text{PPMI}}^{\text{FA}}$, (iii) FastText similarity $\mathbf{S}^{\text{FT}}$, (iv) BERT similarity $\mathbf{S}^{\text{BERT}}$, and (v) cross-model consensus similarity $\mathbf{S}_{\text{m}}^{\text{X}}$. We denote a generic reference geometry by $\mathbf{S}^{\text{ref}}$, where $\mathbf{S}^{\text{ref}} \in \{\mathbf{S}_{\text{PPMI}}^{\text{FC}}, \mathbf{S}_{\text{PPMI}}^{\text{FA}}, \mathbf{S}^{\text{FT}}, \mathbf{S}^{\text{BERT}}, \mathbf{S}_{\text{m}}^{\text{X}}\}$. We sample $n$ = 500,000 word pairs for RSA estimation.

Hidden-state similarities are computed as cosine similarity between layerwise word vectors extracted at layer $\ell$. Behavioral similarity matrices are computed as cosine similarity between cue vectors derived from the cue–response count matrices, using PPMI weighting to correct for frequency effects. Lexical baseline similarities (FastText and BERT) are likewise computed using cosine similarity over the corresponding embedding matrices.

For each layer $\ell$, RSA is performed by vectorizing the upper-triangular entries $(i < j)$ of the hidden-state and reference similarity matrices and computing their Pearson correlation:

$$r_\ell = \text{corr}\Big(\{\mathbf{S}_\ell^{\text{hid}}(i,j)\}_{i<j}, \ \{\mathbf{S}^{\text{ref}}(i,j)\}_{i<j}\Big).$$

### 3.6.2. NEAREST-NEIGHBOR OVERLAP ANALYSIS

As a complementary, local measure, we quantify how well the *nearest-neighbor neighborhoods* induced by hidden-state similarity match those of behavioral and reference spaces (Schnabel et al., 2015). For each model, extraction strategy, and layer $\ell$, we define the $k$-nearest-neighbor *index set* of word $w_i$ under a similarity matrix $\mathbf{S} \in \mathbb{R}^{|\mathcal{V}| \times |\mathcal{V}|}$ as

$$N_k^{\mathbf{S}}(i) := \underset{j \in \{1,\ldots,|\mathcal{V}|\} \setminus \{i\}}{\arg\text{topk}} \mathbf{S}(i,j),$$

i.e., the set of $k$ indices $j \neq i$ with the largest similarities $\mathbf{S}(i,j)$ (ties, if any, are broken deterministically). We then compute the per-word neighborhood overlap between hidden-state similarity and a reference geometry as

$$\text{NN@}k^{(\ell)}(i; \mathbf{S}^{\text{ref}}) = \frac{\left| N_k^{\mathbf{S}_\ell^{\text{hid}}}(i) \cap N_k^{\mathbf{S}^{\text{ref}}}(i) \right|}{k}.$$

We evaluate $k \in \{5, 10, 20, 50, 100, 200\}$ against $\mathbf{S}_{\text{PPMI}}^{\text{FC}}$, $\mathbf{S}_{\text{PPMI}}^{\text{FA}}$, $\mathbf{S}^{\text{FT}}$, $\mathbf{S}^{\text{BERT}}$, and $\mathbf{S}_{\text{m}}^{\text{X}}$. We use the full similarity matrix for nearest-neighbor analyses.

### 3.6.3. HELD-OUT-WORDS RIDGE REGRESSION

We test predictive alignment under explicit generalization constraints by predicting a model's hidden-state similarity from five scalar similarity predictors. Fix a target model

$m$, extraction prompt $s$, and layer $\ell \geq 1$. For each unordered word pair $(i, j)$, we define the regression target and predictors as:

$$y_{ij}^{(m,s,\ell)} := \mathbf{S}_{m,s,\ell}^{\text{hid}}(i,j), \qquad \mathbf{x}_{ij} := \begin{bmatrix} \mathbf{S}^{\text{FT}}(i,j) \\ \mathbf{S}^{\text{BERT}}(i,j) \\ \mathbf{S}_{\text{m}}^{\text{X}}(i,j) \\ \mathbf{S}_{\text{counts}}^{\text{FC}}(i,j) \\ \mathbf{S}_{\text{counts}}^{\text{FA}}(i,j) \end{bmatrix}.$$

Here $y_{ij}^{(m,s,\ell)}$ is the mean-centered cosine similarity between the layer-$\ell$ hidden-state word vectors of $w_i$ and $w_j$, where mean-centering is performed per model/prompt/layer using *training words only* before cosine similarities are computed. The predictors are cosine similarities from FastText, BERT, and the cross-model consensus reference, plus two behavioral similarities computed from raw cue–response counts for FC and FA.

We use raw-count behavioral similarities as regression predictors to avoid leakage: PPMI reweighting depends on global corpus-level marginals (row/column totals), which would otherwise be estimated using test-word counts. The consensus term $\mathbf{S}_{\text{m}}^{\text{X}}(i,j)$ is computed by averaging mean-centered hidden-state cosine similarities over *all layers* (excluding layer 0) of *all other models* $m' \neq m$.

To avoid leakage, we split the vocabulary into 80% training words and 20% test words and form word pairs only within each split (Elangovan et al., 2021). The centering statistics for hidden states (per layer) are computed from the training split and then applied to both training and test words prior to computing $\mathbf{S}_{m,s,\ell}^{\text{hid}}$.

We fit on $n = 100{,}000$ sampled training pairs and evaluate on all available $n = 499{,}500$ test pairs. For each layer $\ell$, we fit a ridge regression with standardized predictors,

$$\hat{\boldsymbol{\beta}}^{(\ell)} = \arg\min_{\boldsymbol{\beta}} \left\| \mathbf{y}^{(\ell)} - \mathbf{X}\boldsymbol{\beta} \right\|_2^2 + \alpha \|\boldsymbol{\beta}\|_2^2,$$

selecting $\alpha$ via 5-fold cross-validation over 15 log-spaced values in $[10^{-2}, 10^6]$, and report test-set $R^2$ as well as incremental gains from adding behavioral predictors (FC/FA) beyond the baseline ($\mathbf{S}^{\text{FT}}, \mathbf{S}^{\text{BERT}}, \mathbf{S}_{\text{m}}^{\text{X}}$).

### 3.7. Data Availability

For reproducibility, all code, including the scripts to generate the LLM association behavior dataset, is available at https://github.com/schiekiera/llm-association-geometry. The full behavioral dataset (17.5M+ trials) is publicly released at https://huggingface.co/datasets/schiekiera/llm-association-geometry.

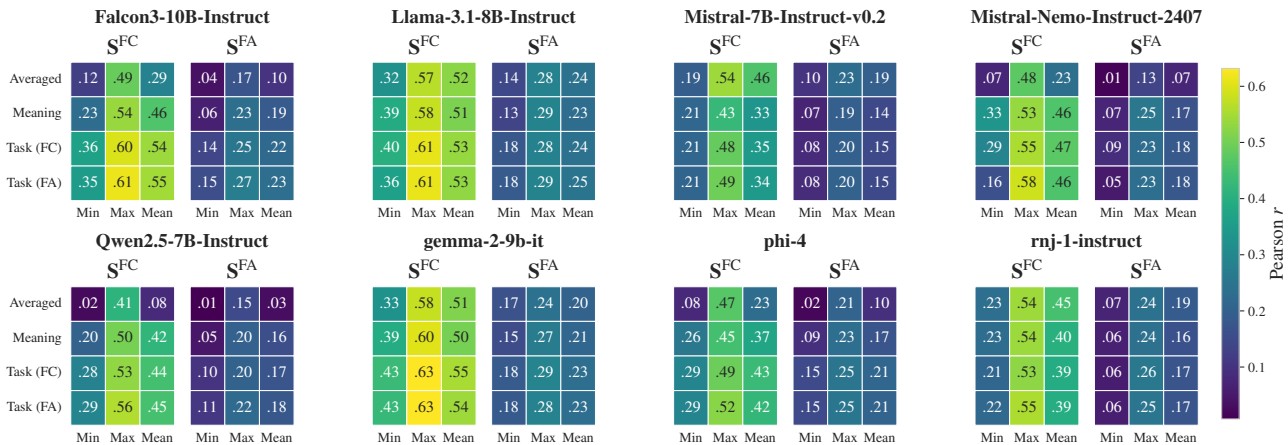

*Figure 4.* Representational similarity analysis between model hidden-state similarity and behavior-derived semantic geometries. Each panel corresponds to a model and contains two sub-heatmaps comparing hidden-state similarity to PPMI-weighted forced-choice ($\mathbf{S}^{FC}$, left) and PPMI-weighted free-association ($\mathbf{S}^{FA}$, right) behavioral embeddings. Rows indicate the embedding extraction strategy (Averaged, Meaning, Task (FC), Task (FA)), and columns indicate layerwise correlations (min, max, mean across layers).

## 4. Results

Three findings organize our results.

- **FC tracks hidden-state geometry more closely than FA.** This holds across global (RSA), local (nearest-neighbor overlap), and predictive (held-out-words ridge regression) measures.

- **FC carries model-specific structure.** FC similarity predicts hidden-state similarity beyond what lexical and cross-model consensus baselines already capture, so it reflects more than the geometry shared across LLMs.

- **The embedding extraction strategy shifts *where* behavior best matches activations.** Meaning-focused prompts peak at early-to-mid layers, while the Averaged strategy moves the alignment peak later in the network.

We report aggregate results in the main text and leave per-model breakdowns to Appendix F.

### 4.1. Representational Similarity Analysis

Figure 4 summarizes RSA results across models and embedding-extraction strategies, while Figure 3a reports layerwise RSA correlations averaged across all models for each reference geometry. Figure 3b further breaks this down by showing the layerwise RSA profiles for the FC and FA reference spaces under each extraction strategy. Averaged across the four extraction strategies, all layers, and all eight models, the mean Pearson correlations are: cross-model consensus $r = .74$, FC $r = .43$, FastText $r = .20$, FA $r = .18$, and BERT $r = .11$. FC, using only discrete text outputs,

therefore recovers roughly $58\%$ of cross-model consensus's alignment with the target model despite having no access to hidden states, logits, or other-model weights.

#### 4.1.1. BEHAVIORAL RSA BY EXTRACTION STRATEGY

Here we examine how the choice of extraction strategy modulates RSA for each behavioral reference. Mean FC RSA increases from $r = .35$ under Averaged to $r = .46$ under Task (FC) and $r = .46$ under Task (FA), with Meaning close at $r = .43$. FA geometry shows the same pattern at lower magnitude ($r = .14$ under Averaged vs. $r = .20$ under task-aligned strategies; Meaning: $r = .18$). More detailed results for low-dimensional projections of the behavioral geometry can be found in Appendix F.1.

#### 4.1.2. BASELINE RSA BY EXTRACTION STRATEGY

As a control, we ask whether the same strategy effect appears for the non-behavioral reference geometries. Each baseline also aligns more strongly under the meaning-focused strategies than under Averaged, though the magnitudes differ. FastText increases from $r = .15$ (Averaged) to $r = .21$ under the other strategies, BERT improves from $r = .08$ to $r = .12$, and cross-model consensus shifts most sharply (mean $r = .57$ under Averaged vs. $r = .79-.80$ under the other strategies).

#### 4.1.3. RSA BY FREQUENCY AND SEMANTIC CATEGORY

To characterize *where* FC adds signal beyond the lexical and cross-model baselines, we re-compute RSA after stratifying the vocabulary by WordNet supersense category and SUBTLEX log-frequency tertile (Appendix F.1). FC outperforms FA in 20 of 21 supersenses, with the largest gap on abstract categories (`cognition`, `attribute`, `feeling`:

$\Delta r \approx +.30$) and narrower gaps on concrete ones (`plant`, `animal`, `food`: $\Delta r \approx +.13$); FC alignment also scales with cue-word frequency ($r = .37, .44, .47$ from low to high).

### 4.1.4. RSA BY LAYER AND STRATEGY

A consistent pattern is that the embedding extraction strategy systematically shifts *where* in the network behavior best matches activations. Under three of the four strategies (Meaning, Task (FC), Task (FA)), the model's hidden-state geometry aligns with FC most strongly at early-to-mid layers (around $30\%$ network depth on average). The fourth strategy, Averaged, shifts alignment peaks much later (around $72\%$ depth). In early and mid layers, meaning-focused FC alignment is roughly $54\%$ higher than under Averaged ($r = .47$ vs. $.31$). Layerwise alignment trajectories under the three meaning-focused strategies are tightly correlated across reference geometries (mean $r = .85$), while they decouple under Averaged (mean $r = .37$), consistent with averaging over many natural contexts diluting the 'word-in-focus' signal by mixing senses and topics (Bommasani et al., 2020; Chronis & Erk, 2020; Derby et al., 2021; Tikhomirova & Wulff, 2026). Detailed per-model layerwise profiles are in Appendix F.1.

### 4.2. Nearest-Neighbor Overlap Analysis

Figure 3a (right) summarizes nearest-neighbor consistency (NN@$k$) between hidden-state similarity and each reference geometry. Across $k$, FC paradigm behavior (NN$_{\text{PPMI}}^{\text{FC}}$) shows the highest agreement among the behavioral embeddings and increases steadily with neighborhood size (.20 at $k = 5$ to .29 at $k = 200$), while FA behavior (NN$_{\text{PPMI}}^{\text{FA}}$) peaks at small neighborhoods (best $k = 5$, .18).

Lexical baselines also improve with larger $k$ (FastText: $.15 \rightarrow .21$; BERT: $.16 \rightarrow .19$), and cross-model consensus yields substantially larger overlap ($.51 \rightarrow .56$), reflecting shared nearest-neighbor structure across models. More detailed results can be found in Appendix F.2.

### 4.3. Held-Out-Words Ridge Regression

Held-Out-words ridge regression shows that cross-model consensus is a highly informative predictor of a target model's hidden-state similarity, with target model behavior providing modest but systematic additional signal. The results of the regression are summarized in Figure 5. Averaged across all model–strategy conditions, adding behavioral FC similarity on top of baseline improves mean test $R^2$ by $+.022$, whereas FA yields a smaller gain ($+.002$); the full model reaches mean $R^2 = .587$ (vs. $.569$ for baseline).

Behavioral gains are largest under the Averaged strategy for several models (e.g., `gemma-2-9b-it`: $+.159$). Peak per-

formance is achieved by `Llama-3.1-8B-Instruct` under Meaning ($R^2 = .844$), and is similarly high for `phi-4` under Task (FC) (.824) and Task (FA) (.817). More detailed results are reported in Appendix F.3, and an ablation study on non-mean-centered hidden-states is reported in Appendix F.4.

## 5. Discussion

We investigated whether an LLM's hidden-state semantic geometry can be recovered from its observable behavior in classic psycholinguistic paradigms, using eight instruction-tuned transformers, a shared 5,000-word noun vocabulary, and 17.5M+ total trials. Behavioral geometry was constructed from cue–response matrices, and compared to layerwise hidden-state similarity (Kriegeskorte et al., 2008; Nili et al., 2014).

Across models and evaluations, FC aligns substantially more with hidden-state geometry than FA. Held-out-words ridge regression further shows that FC contributes signal beyond what lexical and cross-model consensus baselines already capture, indicating that FC carries model-specific structure on top of the shared cross-LLM geometry. Another important effect is the strength of cross-model consensus: similarity structure shared across other LLMs explains a large fraction of variance in a target model's hidden-state geometry, consistent with the assumption of a substantial common semantic subspace (Huh et al., 2024; Kaushik et al., 2025). Furthermore, the choice of hidden-state extraction strategy is itself a substantial moderator of *where* this alignment is highest in the network: under meaning-focused prompts (Meaning, Task (FC), Task (FA)) FC alignment peaks at early-to-mid layers, while averaging hidden states over many natural contexts shifts the peak later.

### 5.1. Implications

Our results have implications for both cognitive science and representation learning research.

For cognitive science, our fully observable language-model setup lets us test a core assumption empirically: that structured behavior is constrained by internal states and can therefore partially reveal them (Baker et al., 2009). The RSA and regression results for FC indicate that behavior-only observations preserve a nontrivial projection of the model's hidden-state similarity geometry. FC has a favorable characteristic here: its controlled candidate sets concentrate observations, producing a less sparse cue–response matrix (De Deyne et al., 2012; Roads & Love, 2021). FA imposes less constraint on the response space (Zemla & Austerweil, 2018; De Deyne et al., 2019), so its cue–response matrix is sparser and more heavy-tailed, with fewer shared columns across cues. Under cosine similarity, this pushes pairwise

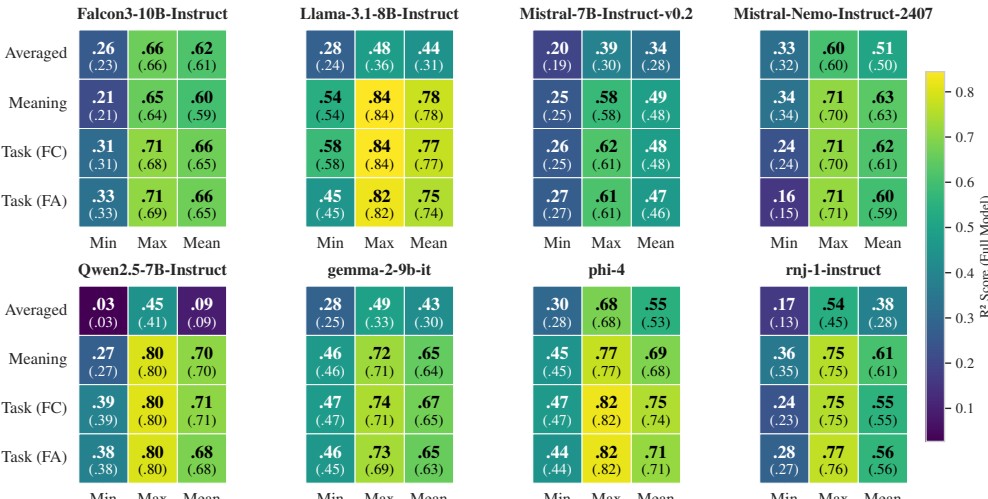

*Figure 5.* Ridge regression performance for predicting hidden-state similarity from behavioral and lexical features across eight models. Bold values show $R^2$ for the full model (behavioral+FastText+BERT+cross-model consensus); parenthetical values show the FastText+BERT+cross-model consensus baseline. Rows indicate the embedding extraction strategy (Averaged, Meaning, Task (FC), Task (FA)), and columns indicate layerwise correlations (min, max, mean across layers).

comparisons toward small intersections, lowering the signal-to-noise ratio for recovering geometric structure. Whether a behavioral task reveals internal structure is therefore not a generic property of 'behavior': protocols that concentrate responses onto shared supports and enforce explicit comparisons (FC) yield higher signal-to-noise measurements of semantic geometry than open-ended production tasks (FA), which disperse probability mass.

For representation learning research, the relevant implication concerns measurement access. The FC protocol acts as a black-box readout of internal semantic geometry: it requires only discrete text outputs, with no access to logits, hidden states, or weights. This contrasts with representational alignment methods that depend on internal-state access (Kornblith et al., 2019; Klabunde et al., 2025; Sucholutsky et al., 2024) and with extraction techniques that rely on logit leakage (Carlini et al., 2024; Gharami et al., 2025). The readout recovers a substantial fraction of cross-model consensus's alignment with the target model without any internal access to it. Because the open-weight setup lets us verify the readout against ground-truth activations, the same protocol applies in principle to closed-source models where activations are inaccessible by construction.

### 5.2. Limitations and Future Directions

Key limitations follow from the observation model and the scope of the evaluation. First, our vocabulary is restricted to high-frequency English nouns, which limits conclusions about other parts of speech, and multilingual semantics. Second, FC results depend on the candidate-set construction (set size, shuffling scheme), which can shape overlap statis-

tics and therefore the stability of the induced cue–response geometry. Finally, our analyses are correlational, so even strong alignment does not by itself establish that particular hidden-state features cause the observed behavior.

Several extensions would broaden coverage. On the measurement side, future work should also compare behavioral geometry from other psycholinguistic paradigms with the hidden-state geometry of LLMs (e.g., rankings, triadic comparisons, best–worst scaling). On the mechanism side, alignment claims would be stronger with causal tests—e.g., directly modifying or removing specific internal activations and checking whether the model's FC/FA similarity structure shifts in the predicted way. Finally, generality can be assessed by expanding beyond nouns and English.

### 5.3. Conclusion

Across eight instruction-tuned LLMs, large-scale behavioral probing recovers meaningful structure in hidden-state semantic geometry, but the fidelity depends strongly on the measurement channel. Forced-choice behavior provides substantially stronger and more reliable alignment than free association, and embedding extraction strategy determines which layers show peak correspondence. Overall, behavioral tasks can reveal aspects of hidden semantic organization when treated as carefully engineered measurement instruments. Constrained comparisons (FC) are a practical lever for increasing recoverability, while open-ended association (FA) appears too noisy to add much signal beyond shared cross-model structure.

## Acknowledgements

This research was partially supported by the Deutsche Forschungsgemeinschaft (DFG) through the projects "What's in a name? Computational modeling and experimental investigations on the non-arbitrariness of word label choices" (project number 459717703), "A computational implementation of the Swinging Lexical Network model of language production" (project number 532390335), the DFG Cluster of Excellence MATH+ (EXC-2046/1, project id 390685689), as well as by the German Federal Ministry of Research, Technology and Space (research campus Modal, fund number 05M14ZAM, 05M20ZBM) and the VDI/VDE Innovation + Technik GmbH (fund number 16IS23025B).

## Impact Statement

This paper studies whether large-scale behavioral probing can recover aspects of hidden-state semantic geometry in LLMs, to improve interpretability and measurement. Potential benefits include stronger evaluation tools, clearer links between behavioral probes and internal representations, and reusable data/code for reproducible research. Potential risks include behavioral "fingerprinting" of models or facilitating imitation when combined with other signals; we mitigate this by using behavior-only discrete outputs (no logits) and analyzing similarity structure rather than reconstructing parameters. We do not foresee direct deployment harms, but note that safeguards may be needed if such methods are used for model auditing or access control.

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

# A. Glossary

## A.1. Abbreviations.

- **LLM**: large language model.

- **RSA**: representational similarity analysis.

- **FC**: forced choice (similarity-based forced-choice paradigm).

- **FA**: free association (free-response association paradigm).

- **PPMI**: positive pointwise mutual information (reweighting of cue–response counts).

- **SVD**: singular value decomposition (used for low-rank variants of behavioral geometry).

- **MLP**: multi-layer perceptron (the feed-forward submodule in a transformer block).

- **C4**: Colossal Clean Crawled Corpus (source of natural contexts for the Averaged strategy).

- **SUBTLEX-US**: English word frequency lexicon used to seed the vocabulary.

## A.2. Paper-Specific Notation.

- $\mathcal{V}$: shared noun vocabulary ($|\mathcal{V}| = 5{,}000$); $w_i$ denotes the $i$-th word.

- $\mathbf{B}$: cue–response count matrix (rows = cues in $\mathcal{V}$; columns = response types).

- $\mathbf{B}^{\mathrm{FC}}$, $\mathbf{B}^{\mathrm{FA}}$: cue–response matrices from forced choice and free association, respectively.

- $\widetilde{\mathbf{B}}$: reweighted cue–response matrix (e.g., via PPMI).

- $\mathbf{S}$: similarity matrix over $\mathcal{V}$ (pairwise cue–cue similarities).

- $\mathbf{S}^{\mathrm{FC}}$, $\mathbf{S}^{\mathrm{FA}}$: behavioral similarity matrices induced by cosine similarity between rows of $\widetilde{\mathbf{B}}^{\mathrm{FC}}$ / $\widetilde{\mathbf{B}}^{\mathrm{FA}}$.

- $\mathbf{S}_\ell^{\mathrm{hid}}$: hidden-state similarity matrix at transformer layer $\ell$ (cosine similarity between extracted word vectors).

- $\mathbf{S}^{\mathrm{FT}}$, $\mathbf{S}^{\mathrm{BERT}}$: FastText and BERT similarity baselines.

- $\mathbf{S}_{\mathrm{m}}^{\mathrm{X}}$: cross-model consensus similarity matrix for target model $m$ (computed from other models).

- $h_\ell(w, c)$: residual-stream hidden states of word $w$ in context $c$ after transformer block $\ell$.

- $\mathbf{e}_\ell^s(w)$: extracted representation of word $w$ at layer $\ell$ under extraction strategy $s \in \{\text{Averaged, Meaning, Task (FC), Task (FA)}\}$.

- $r_\ell$: RSA correlation at layer $\ell$ between upper-triangular entries of $\mathbf{S}_\ell^{\mathrm{hid}}$ and a reference similarity matrix.

- NN@$k$: nearest-neighbor overlap at neighborhood size $k$.

- $N_k^{\mathbf{S}}(i)$: index set of the $k$ nearest neighbors of $w_i$ under similarity matrix $\mathbf{S}$.

- $y_{ij}$, $\mathbf{x}_{ij}$: regression target (hidden similarity) and predictor vector (similarity features) for word pair $(i, j)$.

# B. Models and Identifiers

Table 3 lists the eight instruction-tuned decoder models we used in this study.

*Table 3.* Models used in this study. *Params* = number of parameters in billions; $L$ = number of layers; $d_{\text{model}}$ = hidden dimension size.

| Model ID (Hugging Face) | Citation | Params | L | $d_{\text{model}}$ |
|---|---|---|---|---|
| `tiiuae/Falcon3-10B-Instruct` | (TII Team, 2024) | 10B | 40 | 3072 |
| `google/gemma-2-9b-it` | (Gemma Team, 2024) | 9B | 42 | 3584 |
| `meta-llama/Meta-Llama-3.1-8B-Instruct` | (Meta AI, 2024) | 8B | 32 | 4096 |
| `mistralai/Mistral-7B-Instruct-v0.2` | (Jiang et al., 2023) | 7B | 32 | 4096 |
| `mistralai/Mistral-Nemo-Instruct-2407` | (Mistral AI, 2024) | 12B | 40 | 5120 |
| `microsoft/phi-4` | (Abdin et al., 2024) | 14B | 40 | 5120 |
| `Qwen/Qwen2.5-7B-Instruct` | (Qwen Team, 2024) | 7B | 28 | 3584 |
| `EssentialAI/rnj-1-instruct` | (Vaswani et al., 2025) | 8B | 32 | 4096 |

# C. Preprocessing

## C.1. Vocabulary Construction and C4 Sentence Retrieval

**Filtering.** Starting from SUBTLEX-US, we: (i) keep only rows with `Dom_PoS_SUBTLEX == "Noun"`, (ii) remove a fixed list of contraction fragments (e.g., `isn`, `aren`, `ll`, `re`, etc.), (iii) lemmatize with spaCy (`en_core_web_sm`) and deduplicate by lemma, keeping the most frequent row, (iv) drop non-string entries and words with length $\leq 2$, and (v) select the top 6,000 by SUBTLEX frequency.

**C4 retrieval.** We stream the C4 English split and collect a maximum of 500 sentences per word, filtering sentences by length (5–100 whitespace tokens) and matching by simple alphanumeric tokenization. We keep the 5,000 highest-frequency words that have at least 50 collected sentences and downsample to exactly 50 sentences per word. These 50 sentences define the contexts used by the `averaged` hidden-state extraction strategy.

## C.2. Benchmark Embeddings (FastText and BERT)

**FastText.** We load English FastText vectors from `cc.en.300.vec.gz` (Common Crawl), align them case-insensitively to the vocabulary, and compute cosine similarities (Bojanowski et al., 2017).

- `cc.en.300.vec.gz` (Bojanowski et al., 2017)

**BERT.** We use `bert-base-uncased` from Hugging Face and embed each target word in the base prompt `"This is a "`. We isolate the target word's character span using offset mappings and average the aligned WordPiece token vectors from the final hidden layer. We then compute cosine similarities (Devlin et al., 2019).

- `google-bert/bert-base-uncased` (Devlin et al., 2019)

## C.3. PPMI-Weighted Behavioral Embeddings

For both paradigms, model outputs are aggregated into a sparse behavioral cue–response count matrix $\mathbf{B}$, where rows correspond to cue words and columns correspond to unique response words. We denote the matrix for the forced-choice paradigm as $\mathbf{B}^{\text{FC}}$ and for the free-association paradigm as $\mathbf{B}^{\text{FA}}$. In the next step, we apply positive pointwise mutual information (PPMI) to reweight cue–response co-occurrences.

Concretely, letting $B_{ij}^p$ be the count for cue $w_i$ and response $r_j$ under paradigm $p \in \{\text{FC}, \text{FA}\}$, and $N^p = \sum_{i,j} B_{ij}^p$, we define

$$P^p(i,j) = \frac{B_{ij}^p}{N^p}, \qquad P^p(i) = \sum_j P^p(i,j), \qquad P^p(j) = \sum_i P^p(i,j),$$

$$\text{PMI}^p(i,j) = \log \frac{P^p(i,j)}{P^p(i)\,P^p(j)}, \qquad \text{PPMI}^p(i,j) = \max\big(0, \text{PMI}^p(i,j)\big).$$

We then form the reweighted matrix $\widetilde{\mathbf{B}}^p$ with entries $\widetilde{B}_{ij}^p = \text{PPMI}^p(i,j)$ and compute cue–cue similarities via cosine similarity between rows,

$$\mathbf{S}^p(i,k) = \cos\big(\widetilde{\mathbf{B}}_{i,:}^p, \widetilde{\mathbf{B}}_{k,:}^p\big).$$

## C.4. Hidden-State Extraction: Prompts and Token-Span Isolation

**Extraction prompts.** The four extraction strategies in the main text correspond to:

- `Averaged`: 50 C4 sentences containing $w$

- `Meaning`: `"What is the meaning of the word {w}?"`

- `Task (FC)`: FC-style instruction prompt with the cue inserted (without the candidate list; see Appendix D.1 for the full prompt)

- `Task (FA)`: FA-style instruction prompt with the cue inserted (see Appendix E.1 for the full prompt)

**Token span isolation.** For each prompt, we locate the last occurrence of the cue substring and use tokenizer offset mappings to select all non-special tokens whose character spans overlap the cue span; we then average hidden states over the selected positions. For `averaged` we compute these vectors for each of the 50 contexts and average them. We compute cosine similarity matrices for all layers except layer 0 returned by `output_hidden_states` by normalizing word vectors and taking dot products.

# D. Forced-Choice Data Collection

## D.1. Forced-Choice Prompting.

The FC task asks for exactly two selections from a provided candidate list; generation enforces formatting and retries non-compliant outputs. Candidate pools are constructed deterministically: for each cue, the remaining vocabulary is shuffled with a fixed seed and partitioned into groups of at most 16 candidates, yielding one FC trial per group.

**FC behavioral prompt template (verbatim).**

```
You will be given one input word and a list of candidate words.
Your task is to select exactly {n_picks} words from the list that are most
similar or closely related to the input word.

Rules:
- Select exactly {n_picks} words.
- Both selected words must come from the provided candidate list.
- Do not select the input word.
- Output must contain only the {n_picks} chosen words.
- Use the format: output: word1, word2
- Do not add any explanation, reasoning, commentary, or extra text.
- Do not change spelling or number of words.

Example:
input word: dog
candidates: [banana, violin, therapy, beer, tango, paper, cat, kiwi,
             jeans, car, vacation, note, leash, bath, ceiling, ivy]
output: cat, leash

Now follow the same format.

input word: {input_word}
candidates: [{candidate_list}]
output:
```

## D.2. Forced-Choice Data Collection Pipeline.

For each cue word, we deterministically constructed candidate sets of size $\leq 16$ by shuffling the remaining vocabulary with a cue-specific seed and partitioning it into balanced groups, yielding 313 trials per cue. Generation was run in batches of 128 prompts with a maximum of 10 newly generated tokens per prompt, using deterministic decoding (`do_sample=False`) and model-specific end-of-turn terminators. If an output was non-compliant (e.g., wrong format or choices outside the

*Table 4.* Forced choice: Summary statistics for compliance and repair across models.

| Model abbreviation | Initial compliance (%) | Final compliance (%) |
|---|---|---|
| All models | 85.3 | 96.2 |
| Falcon3-10B-Instruct | 89.1 | 96.3 |
| gemma2-9b-it | 90.1 | 96.3 |
| Llama3.1-8B-Instruct | 79.0 | 96.3 |
| Mistral7B-Instruct-v0.2 | 81.8 | 86.8 |
| MistralNemo-Instruct-2407 | 92.9 | 98.8 |
| phi-4 | 96.9 | 98.8 |
| Qwen2.5-7B-Instruct | 90.5 | 97.1 |
| rnj1-instruct | 62.3 | 99.0 |

candidate set), we issued an explicit repair prompt; remaining failures were re-prompted with up to five sampled retries using nucleus sampling ($T = 0.5$, top-$p = 0.9$), with deterministic seeding for reproducibility.

### D.3. Forced Choice Prompt Compliance Analysis

To maximize usable trials, we applied an automated compliance-and-retry procedure during data collection. After an initial deterministic generation pass, each response was checked for compliance (i.e., exactly two selections, both drawn from the provided candidate list, and excluding the cue word). Non-compliant outputs triggered a deterministic repair prompt that restated the rules and flagged the previous answer as invalid; if the model still failed, we issued up to 5 additional retry prompts using stochastic decoding ($temperature = 0.5$, $top - p = 0.9$). All prompts and retries were executed in batches, and the final output per trial was the last compliant response obtained (or, if no retry succeeded, the last generated response was retained and filtered out during postprocessing).

An overview of the prompt compliance and repair effectiveness across models is shown in Table 4 and over usable associations in Table 6. Initial compliance ranged from 62.3% (rnj1-instruct) to 96.9% (phi-4), with most models clustered around 80%-93%. After applying the repair and retry procedures, final compliance increased to 96.3%-99.0% for seven of eight models, indicating that nearly all trials could be standardized to the target format. The main exception was Mistral7B-Instruct-v0.2, which improved more modestly (from 81.8% to 86.8%), leaving a larger fraction of unusable outputs relative to the other models.

## E. Free-Association Data Collection

### E.1. Free Association Prompting.

The FA task asks for exactly five single-word associations in a single line. For each model we run multiple stochastic generations per cue with different random seeds (in the current pipeline, 126 runs).

**FA behavioral prompt template (verbatim).**

```
You will be given one input word.
Produce exactly five different single-word associations.

Rules:
- Output only five associated words.
- Each must be a single word (no spaces or punctuation inside a word).
- All five words must be different from each other.
- Do not repeat the input word.
- Order the words by how quickly they come to mind (first = strongest).
- Format your answer as a single line starting with 'output:'.
- Separate the five words with commas and a space.
- End the line with a period.
- Do not add any explanations or extra text.
Example:
input: dog.
```

*Table 5.* Free association: Overall quality report for free association outputs. Cue repetition is the percentage of response trials (not associations) that contain the input cue word as an output word. Unique words (total) is the total number of unique words in all responses. $M$ unique per cue is the mean number of unique words per cue.

| Model | Cue repetition (%) | Unique words (total) | $M$ unique per cue |
|---|---|---|---|
| All models | 3.2 | 18,621 | 21.81 |
| gemma-2-9b-it | 0.2 | 12,231 | 14.40 |
| Mistral-Nemo-Instruct-2407 | 3.9 | 18,049 | 24.26 |
| phi-4 | 2.6 | 17,936 | 20.29 |
| rnj-1-instruct | 4.8 | 32,203 | 31.90 |
| Qwen2.5-7B-Instruct | 3.9 | 17,395 | 18.44 |
| Falcon3-10B-Instruct | 2.3 | 17,027 | 19.84 |
| Mistral-7B-Instruct-v0.2 | 0.3 | 18,728 | 19.76 |
| Llama-3.1-8B-Instruct | 7.8 | 15,395 | 25.61 |

*Table 6.* Usable associations from behavioral paradigms. Postprocessing summaries from the cue–response *counts* matrices for forced choice and free association.

| Model | Forced choice | | | Free association | | |
|---|---|---|---|---|---|---|
| | $M$ | % | $\sum$ | $M$ | % | $\sum$ |
| All models | 610.1 | 97.5% | 24,405,587 | 622.6 | 98.8% | 24,905,314 |
| Falcon3-10B-Instruct | 614.0 | 98.1% | 3,070,010 | 625.3 | 99.3% | 3,126,269 |
| Llama-3.1-8B-Instruct | 611.7 | 97.7% | 3,058,606 | 603.8 | 95.8% | 3,019,162 |
| Mistral-7B-Instruct-v0.2 | 562.9 | 89.9% | 2,814,614 | 627.2 | 99.6% | 3,135,754 |
| Mistral-Nemo-Instruct-2407 | 621.1 | 99.2% | 3,105,303 | 624.7 | 99.2% | 3,123,447 |
| Qwen2.5-7B-Instruct | 616.7 | 98.5% | 3,083,390 | 624.3 | 99.1% | 3,121,442 |
| gemma-2-9b-it | 610.3 | 97.5% | 3,051,539 | 629.6 | 99.9% | 3,148,177 |
| phi-4 | 622.0 | 99.4% | 3,109,771 | 626.2 | 99.4% | 3,130,760 |
| rnj-1-instruct | 622.5 | 99.4% | 3,112,354 | 620.1 | 98.4% | 3,100,303 |

```
output: bark, leash, pet, animal, cat.

input: {input_word}
```

### E.2. Free-Association Data Collection Pipeline.

To obtain multiple stochastic samples per cue, we repeated the procedure for $N_{\text{runs}} = 126$ independent runs. Generation used nucleus sampling with temperature $T = 0.7$ and top-$p = 0.95$, with a maximum of 25 newly generated tokens per prompt. Prompts were formatted using each model's chat template (via `apply_chat_template`). For efficiency, cue words were processed in batches of 128 prompts.

### E.3. Free Association Prompt Compliance Analysis

An overview of usable associations per model can be found in Table 6 and information about the cue repetition and unique words per model in Table 5. Overall, 98.8% of associations could be included in the behavioral similarity matrices. Diversity varied substantially across models, with total unique responses ranging from 12,231 (gemma-2-9b-it) to 32,203 (rnj-1-instruct), and mean unique associates per cue spanning $14.40$ to $31.90$, suggesting systematic differences in lexical variety and sampling breadth even under a fixed prompting protocol.

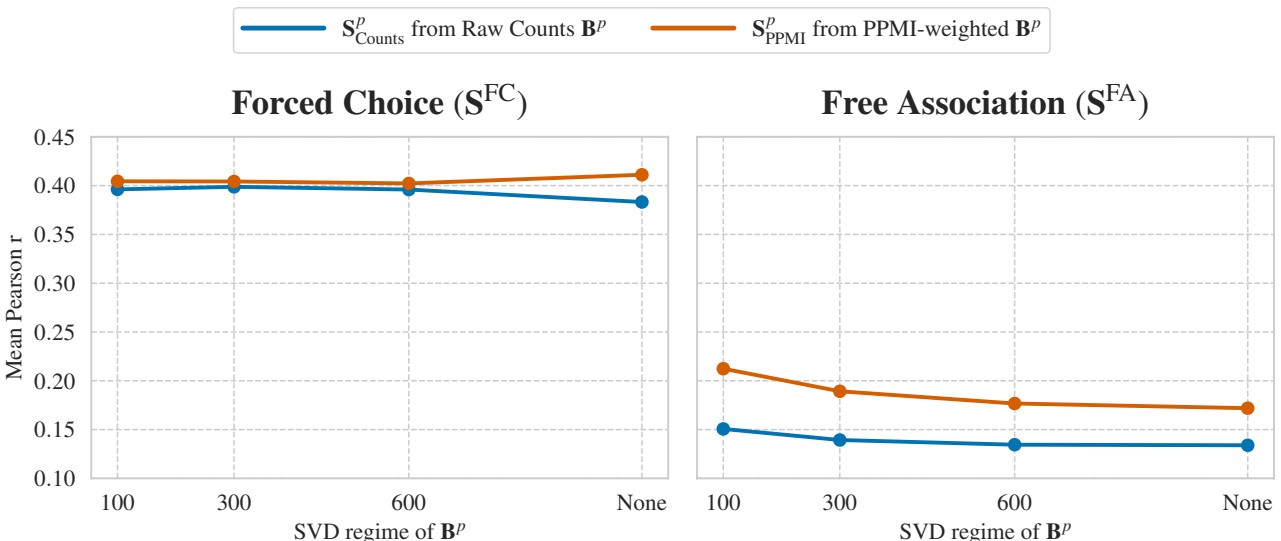

*Figure 6.* Mean Pearson correlation between behavioral semantic spaces and model hidden states as a function of SVD dimensionality reduction applied to behavioral matrices. Left: Forced choice ($S^{\mathrm{FC}}$). Right: Free association ($S^{\mathrm{FA}}$). Blue: raw co-occurrence counts; Orange: PPMI-weighted counts.

# F. Detailed Results

## F.1. Representational Similarity Analysis

### Low-Dimensional Projections of Behavioral Geometry.

Furthermore, we assess the robustness of behavior–activation alignment to alternative constructions of the behavioral geometry. Let $\mathbf{B}^p \in \mathbb{R}^{|\mathcal{V}| \times |\mathcal{R}|}$ denote the cue–response count matrix for paradigm $p \in \{\mathrm{FC}, \mathrm{FA}\}$, and let $\widetilde{\mathbf{B}}^p$ denote its reweighted version obtained either by using raw counts ($\widetilde{\mathbf{B}}^p = \mathbf{B}^p$) or by applying PPMI elementwise to yield $\widetilde{\mathbf{B}}^p = \mathrm{PPMI}(\mathbf{B}^p)$. From $\widetilde{\mathbf{B}}^p$ we derive a behavioral similarity matrix $\mathbf{S}^p$ by cosine similarity between cue rows, $\mathbf{S}^p(i,j) = \cos(\widetilde{\mathbf{B}}^p_{i,:}, \widetilde{\mathbf{B}}^p_{j,:})$. In addition, we consider low-rank behavioral geometries obtained via a truncated SVD $\widetilde{\mathbf{B}}^p \approx \mathbf{U}^p_K \mathbf{\Sigma}^p_K (\mathbf{V}^p_K)^\top$ and define cue embeddings $\mathbf{Z}^p_K := \mathbf{U}^p_K \mathbf{\Sigma}^p_K$, inducing $\mathbf{S}^p_K(i,j) = \cos(\mathbf{Z}^p_K[i,:], \mathbf{Z}^p_K[j,:])$. Throughout, we use $K \in \{100, 300, 600\}$. For each layer $\ell$, we then compute Pearson correlations between the upper-triangular entries of $\mathbf{S}^{\mathrm{hid}}_\ell$ and each behavioral variant (counts vs. PPMI, and full-rank vs. low-rank $\mathbf{S}^p_K$), quantifying how sensitive RSA alignment is to frequency reweighting and dimensionality reduction. Figure 6 shows the mean Pearson correlation between behavioral semantic spaces and model hidden states as a function of SVD dimensionality reduction applied to behavioral matrices.

Figure 6 shows that for FC behavior–activation alignment is stable across PPMI reweighting and low-rank SVD. In contrast, FA produces a sparser, heavy-tailed matrix in which alignment improves with PPMI and stronger SVD compression, consistent with denoising that suppresses rare/idiosyncratic responses and increases effective overlap between cue distributions.

### Detailed Plots for RSA.

Figure 7 aggregates results across models: the top row reports mean RSA as a function of layer for each reference geometry. Figure 8 aggregates means of RSA correlations for each model, reference geometry and embedding extraction strategy. Figure 9 provides the full model-by-model RSA profiles, showing how alignment between hidden-state similarity and each reference geometry (FC behavior, FA behavior, FastText, and BERT) varies across layers and embedding extraction strategies. Layerwise, FC alignment peaks early for task-aligned strategies (layers $10-11$) but peaks late under Averaged (layer 42). At the model level, the strongest mean FC RSA is observed for `gemma-2-9b-it` under Task (FC) ($r = .55$), while the weakest is `Qwen2.5-7B-Instruct` under Averaged ($r = .08$).

**Stratified RSA.** To better characterize *where* FC adds signal beyond the shared structure captured by lexical and cross-model baselines, we re-compute RSA after stratifying the 5,000-word vocabulary along two dimensions: WordNet supersense

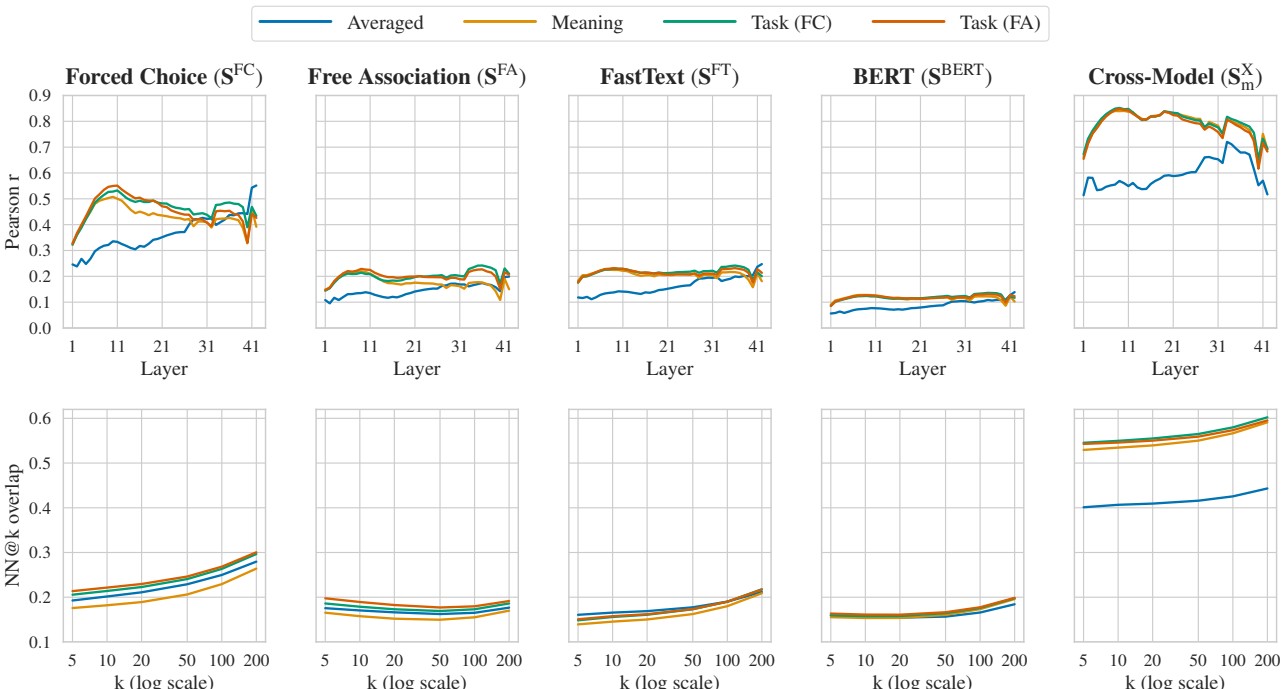

*Figure 7.* Layer-wise representational similarity analysis (top row) and nearest-neighbor consistency (bottom row) between model hidden-state geometry and multiple reference semantic spaces. Columns correspond to reference geometries: PPMI-weighted forced choice ($\mathbf{S}^{\text{FC}}$), PPMI-weighted free association ($\mathbf{S}^{\text{FA}}$), FastText ($\mathbf{S}^{\text{FT}}$), BERT ($\mathbf{S}^{\text{BERT}}$), and cross-model consensus ($\mathbf{S}_{\text{m}}^{\text{X}}$). Top row: Mean Pearson correlation between hidden-state similarity and each reference geometry as a function of transformer layer, averaged across models. Bottom row: Nearest-neighbor overlap (NN@$k$) between hidden states and each reference geometry as a function of neighborhood size $k$ (log-scaled). Colors denote embedding extraction strategies (Averaged, Meaning, Task (FC), Task (FA)).

category and SUBTLEX log-frequency tertile. Within each stratum, we restrict pairwise comparisons to word pairs whose two members both belong to the stratum, then re-compute the Pearson correlation between $\mathbf{S}_{\ell}^{\text{hid}}$ and each behavioral reference geometry. Figure 10 summarizes the result for the ten most populated supersenses and the three frequency tertiles. FC outperforms FA in 20 of 21 supersense categories (the sole exception is `quantity`, where FC and FA are tied at $r \approx .50$). The FC–FA gap is largest for abstract categories (`cognition`: $+.32$, `attribute`: $+.30$, `feeling`: $+.29$) and narrower for concrete categories (`plant`: $+.15$, `animal`: $+.15$, `food`: $+.10$). FC alignment also increases with word frequency ($r = .37, .44, .47$ from low to high), consistent with the higher reliability of behavioral estimates for frequent cues.

### F.2. Nearest-Neighbor Overlap Analysis

In summary, embedding extraction strategy effects mirror RSA. The bottom row of Figure 7 reports the corresponding NN@$k$ trends, enabling a direct comparison of global (RSA) versus local (nearest-neighbor) agreement. At their best $k$, $\text{NN}_{\text{PPMI}}^{\text{FC}}$ is highest under Task (FA)/Task (FC) ($\approx .30$ at $k = 200$) and lowest under Meaning (.26); $\text{NN}_{\text{PPMI}}^{\text{FA}}$ is likewise highest under Task (FA) (.20 at $k = 5$). Cross-model consensus at $k = 200$ is strongest under Task (FC) (.60) and weakest under Averaged (.44). Layerwise, Averaged peaks later (e.g., $\text{NN}_{\text{PPMI}}^{\text{FC}}$ at layer $\sim 22.8$, FastText at $\sim 24.1$), while task-aligned strategies peak earlier (typically $\sim 8$–$12$ for Meaning/Task (FA)/Task (FC)). Model-wise, the best $\text{NN}_{\text{PPMI}}^{\text{FC}}$ is observed for `gemma-2-9b-it` under Task (FC) at $k = 200$ (.36), whereas the lowest overlaps typically occur for `Qwen2.5-7B-Instruct` under Averaged strategies (e.g., $\text{NN}_{\text{PPMI}}^{\text{FC}} = .12$ at $k = 5$; cross-model $= .28$ at $k = 5$).

### F.3. Held-Out-Words Ridge Regression

**Detailed Ridge Results.** Figure 12 shows the incremental gain in held-out-words ridge regression from adding behavioral predictors (FC, FA, and FC+FA) relative to a baseline with lexical and cross-model features, broken down by model. Figure 13 plots the full-model $R^2$ across layers for each model, shown separately for the four hidden-state extraction

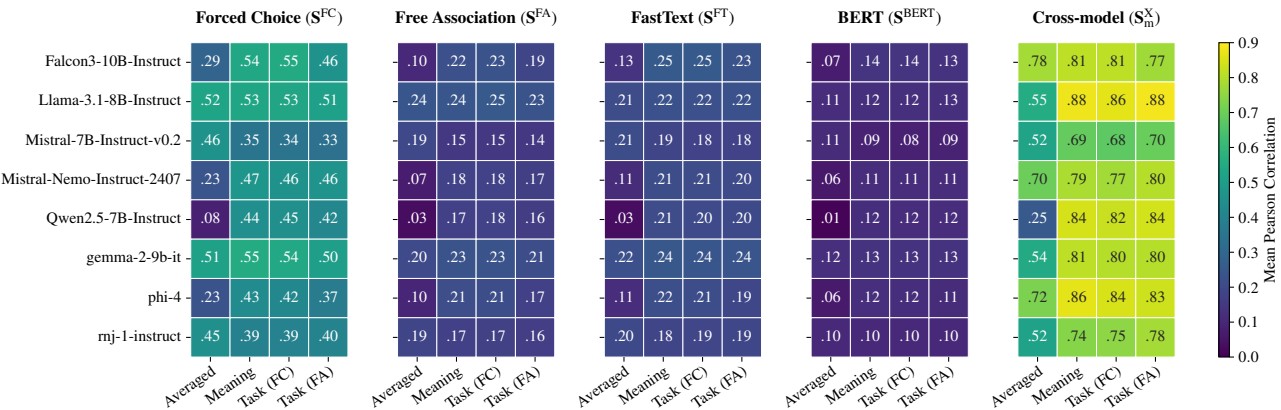

*Figure 8.* Mean RSA (Pearson) between layerwise hidden-state similarity and five reference semantic geometries (PPMI-weighted forced choice, PPMI-weighted free association, FastText, BERT, cross-model consensus). Rows correspond to models and columns to embedding-extraction strategies (Averaged, Meaning, Task (FC), Task (FA)); values are averaged across layers, with color indicating correlation magnitude.

strategies (Averaged, Meaning, Task (FC), Task (FA)).

## F.4. Ablation Study: Ridge Regression with Non-Mean-Centered Hidden States

To assess whether mean-centering is required for our ridge-based RSA mapping, we repeated the full pipeline using raw (non-mean-centered) hidden states when constructing hidden-state cosine-similarity matrices. Results are summarized in Figures 15. Furthermore, Figure 15 reports the incremental gain in held-out-words ridge regression performance from adding behavioral predictors (FC, FA, and FC+FA) relative to a baseline that includes lexical similarity and cross-model consensus features, shown separately for each model, while Figure 16 plots full-model $R^2$ across layers for each model, shown separately for the four hidden-state extraction strategies (Averaged, Meaning, Task (FC), Task (FA)).

Overall, the ridge mapping remains effective without mean-centering (mean $R^2_{\text{baseline}} = .493$; mean $R^2_{\text{full}} = .503$), and the best-performing layers remain in a comparable depth range (mean best layer $\approx 22.3$). However, averaged across all model–prompt settings, the mean-centered pipeline performs better: mean $R^2_{\text{baseline}}$ increases from .493 to .569 ($\Delta = +.076$), and mean $R^2_{\text{full}}$ increases from .503 to .587 ($\Delta = +.084$). A similar advantage is visible in the aggregate peak full-model performance across layers, with mean peak $R^2_{\text{full}}$ rising from .665 (non-mean-centered) to .691 (mean-centered; $\Delta = +.026$).

Importantly, the gain observed for FC is smaller on average in the non-mean-centered condition (mean-centered: $\Delta_{\text{FC}} = .022$; non-mean-centered: $\Delta_{\text{FC}} = .004$) but remains consistently positive across all models (see Figure 15). In contrast, the already small effect for FA in the mean-centered pipeline (mean-centered: $\Delta_{\text{FA}} = .002$) disappears in the non-mean-centered condition (non-mean-centered: $\Delta_{\text{FA}} \approx .000$). In summary, mean-centering yields higher average predictive accuracy and larger explanatory gains for the FC task.

## F.5. Behavior-Only Model Fingerprinting

To quantify how much model-specific structure is carried by FC versus FA behavioral profiles, we train a multinomial logistic-regression classifier to predict which of the eight models a cue's similarity profile comes from. We use a three-way word split to eliminate leakage: 60% train rows (3,000 cues), 20% test rows (1,000 cues), and 20% feature columns (1,000 cues). The features for any cue are its cosine similarities to the 1,000 feature-column cues; the train, test, and feature word sets are pairwise disjoint, so neither the same row nor (by symmetry) the same column is seen at both training and evaluation time. With as few as 100 features, FC profiles already reach 92.1% logistic accuracy across the eight models, and the full 1,000-feature classifier reaches 99.96%. FA profiles top out at 58.7% under the same protocol (permutation chance: 12.5%).

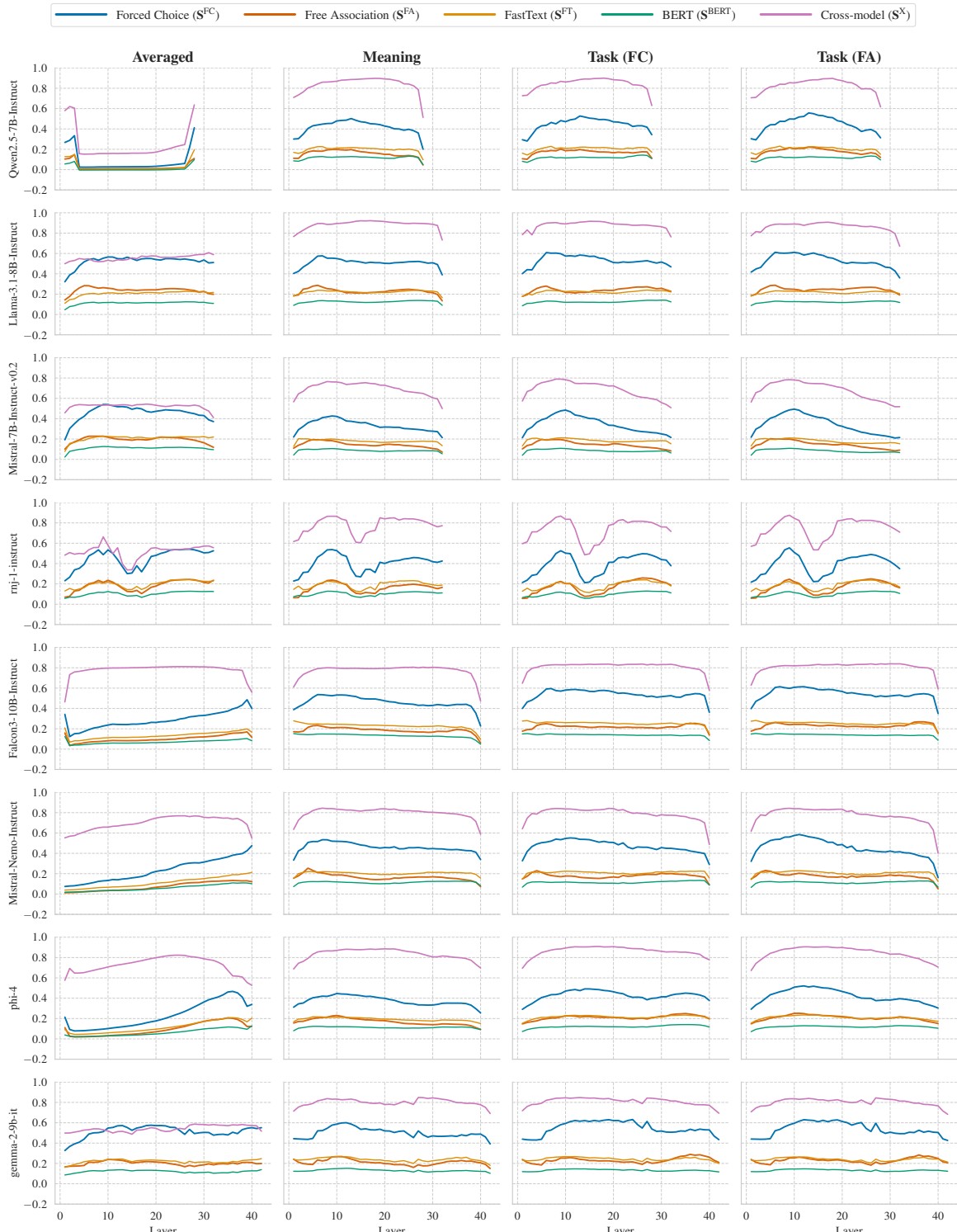

*Figure 9.* Layerwise representational similarity analysis profiles across models and prompting strategies. Rows correspond to instruction-tuned decoder models, and columns correspond to hidden-state extraction strategies: Averaged natural contexts (Averaged), meaning-inducing prompt (Meaning), Task-aligned forced choice prompt (Task (FC)), and Task-aligned free association prompt (Task (FA)). Curves show Pearson correlations between layerwise hidden-state similarity matrices and four reference semantic geometries: PPMI-weighted forced choice ($\mathbf{S}^{FC}$), PPMI-weighted free association ($\mathbf{S}^{FA}$), FastText ($\mathbf{S}^{FT}$), and BERT ($\mathbf{S}^{BERT}$). The x-axis denotes transformer layer index (excluding the embedding layer), and the y-axis denotes RSA correlation.

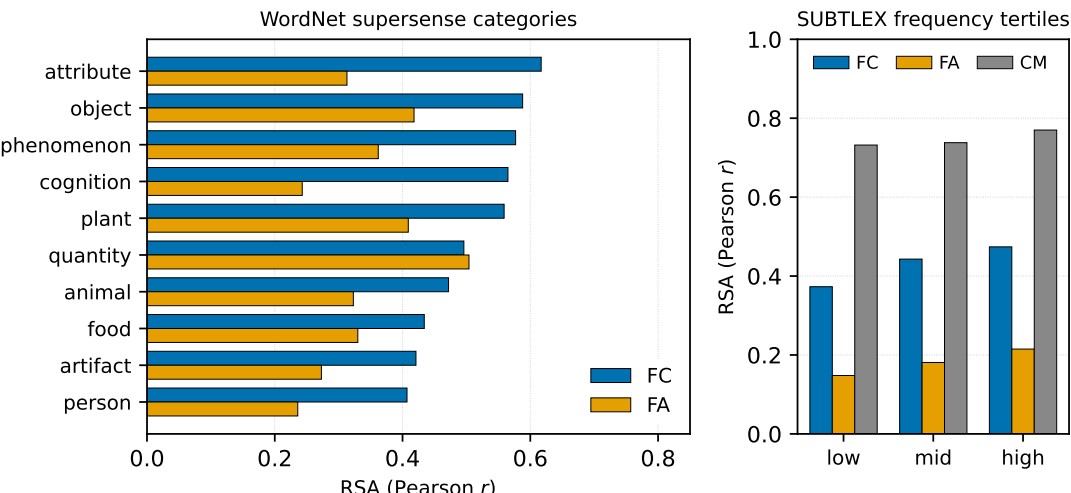

*Figure 10.* Stratified RSA: FC and FA alignment with hidden-state geometry broken down by WordNet supersense category (*left*; top ten categories shown, sorted by FC value) and SUBTLEX log-frequency tertile (*right*; cross-model consensus [CM] shown for reference). Values are mean Pearson $r$ across the eight models, four extraction strategies, and all layers. Per-category vocabulary sizes (left panel, top to bottom): `attribute` = 173, `object` = 88, `phenomenon` = 58, `cognition` = 242, `plant` = 92, `quantity` = 71, `animal` = 149, `food` = 165, `artifact` = 822, `person` = 849. Frequency tertiles (right panel): low = 1670, mid = 1664, high = 1666 words. FC outperforms FA in 20 of 21 supersenses (only `quantity` is tied), with the largest gap on abstract categories. FC alignment scales with cue-word frequency.

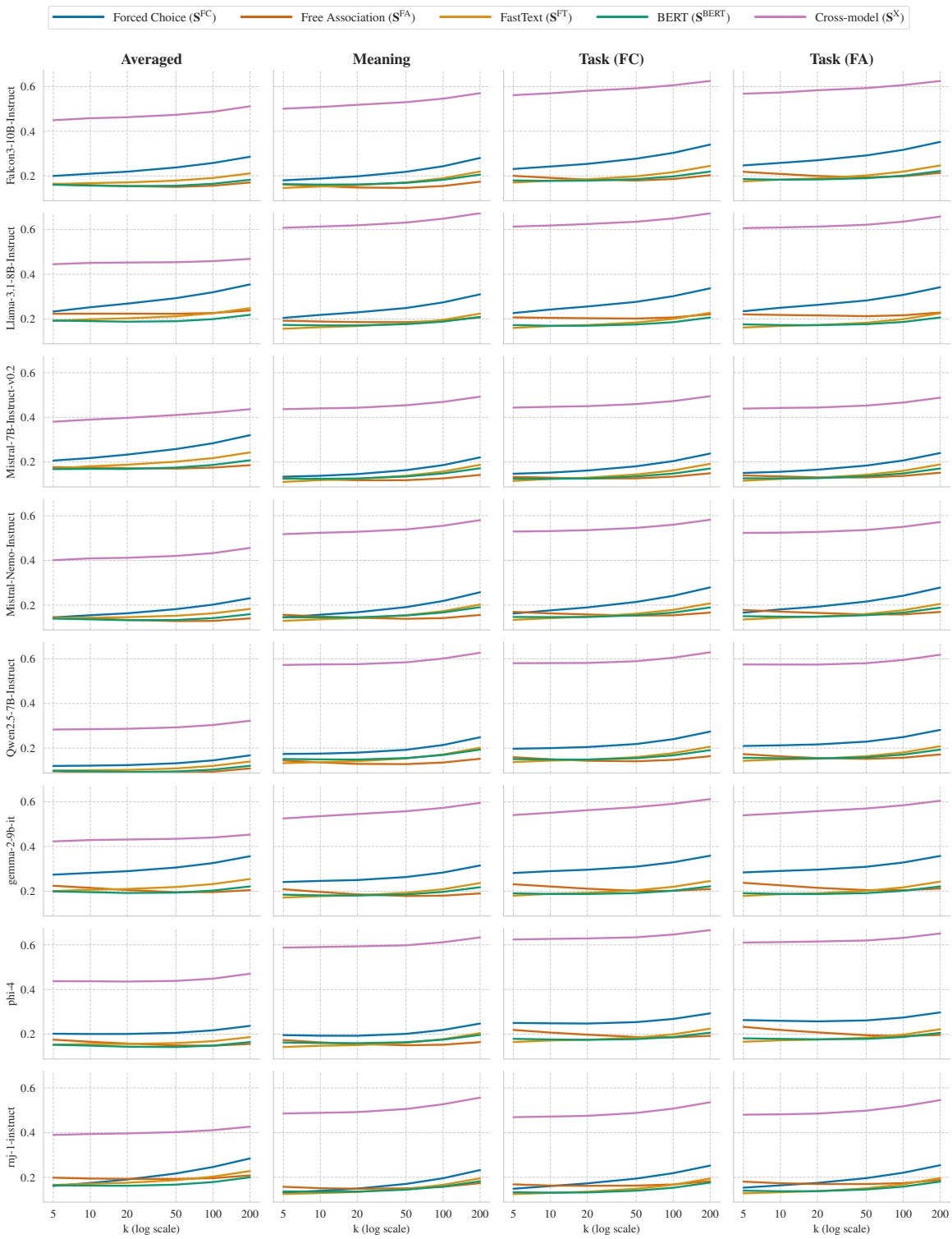

*Figure 11.* Layerwise nearest-neighbor overlap analysis profiles across models and prompting strategies. Rows correspond to instruction-tuned decoder models, and columns correspond to hidden-state extraction strategies: Averaged natural contexts (Averaged), Meaning prompt (Meaning), Task-aligned forced choice prompt (Task (FC)), and Task-aligned free association prompt (Task (FA)). Curves show nearest-neighbor overlap between layerwise hidden-state representations and four reference semantic geometries: PPMI-weighted forced choice ($\mathbf{S}^{FC}$), PPMI-weighted free association ($\mathbf{S}^{FA}$), FastText ($\mathbf{S}^{FT}$), and BERT ($\mathbf{S}^{BERT}$). The x-axis denotes nearest-neighbor neighborhood size $k$ (log-scaled), and the y-axis denotes NN@$k$.

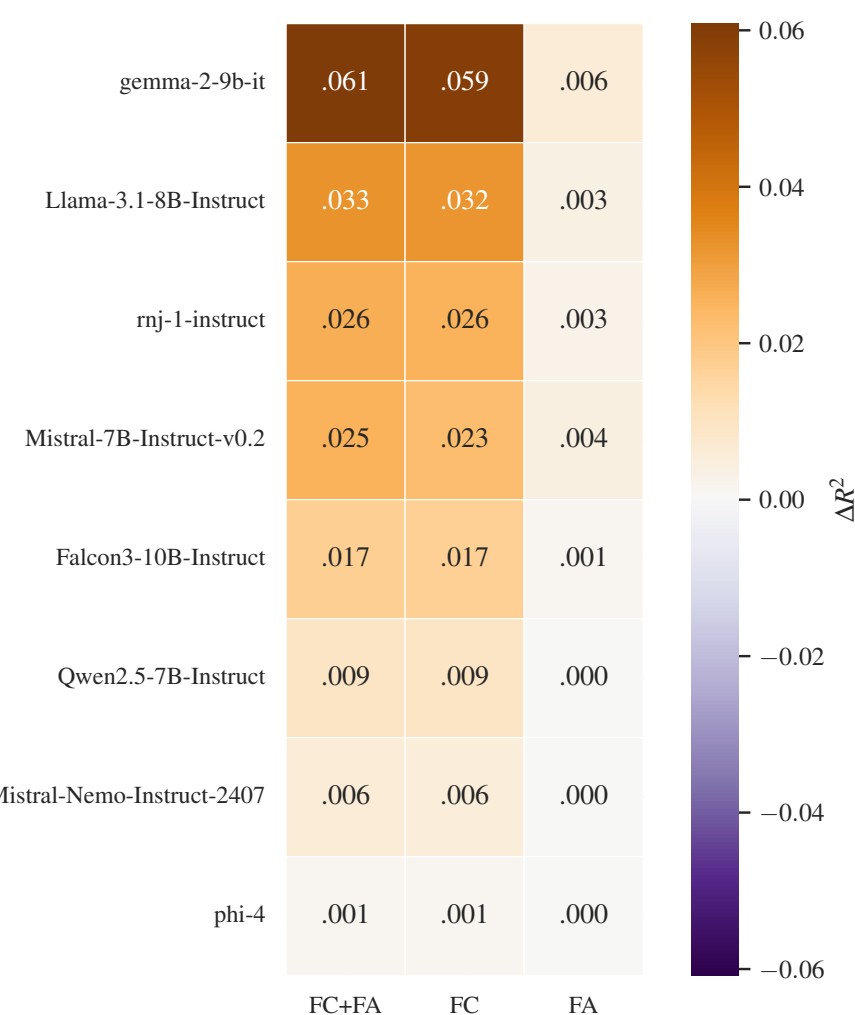

*Figure 12.* Incremental contribution of behavioral predictors to held-out-words ridge regression performance, reported as $\Delta R^2$ relative to a baseline including lexical and cross-model similarity features.

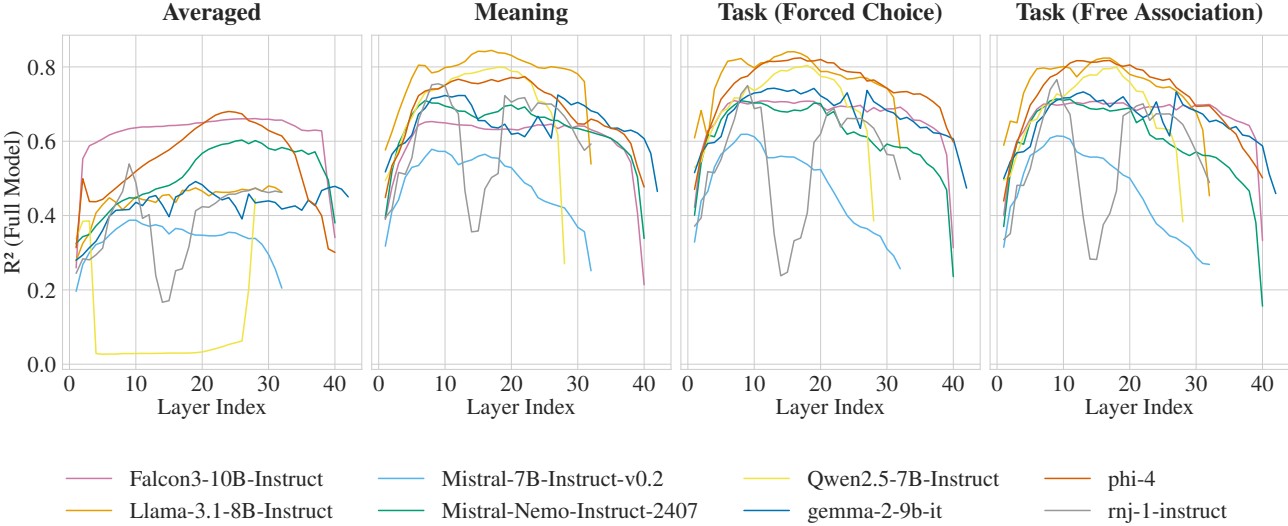

*Figure 13.* Layer-wise held-out-words ridge performance ($R^2$) for predicting each model's hidden-state similarity from behavioral and lexical similarity features, shown separately for the four embedding extraction strategies.

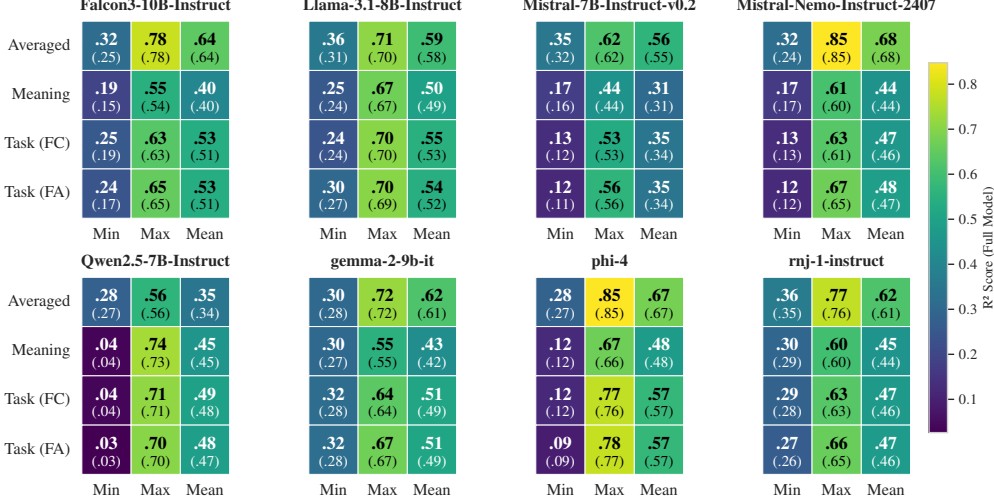

*Figure 14.* Ablation study: Ridge regression performance for predicting non-mean-centered hidden-state similarity from behavioral and lexical features across eight models. Bold values show $R^2$ for the full model (behavioral + FastText+BERT+cross-model consensus); parenthetical values show the FastText+BERT+cross-model consensus baseline. Rows indicate the embedding extraction strategy (Averaged, Meaning, Task (FC), Task (FA)), and columns indicate layerwise correlations (min, max, mean across layers).

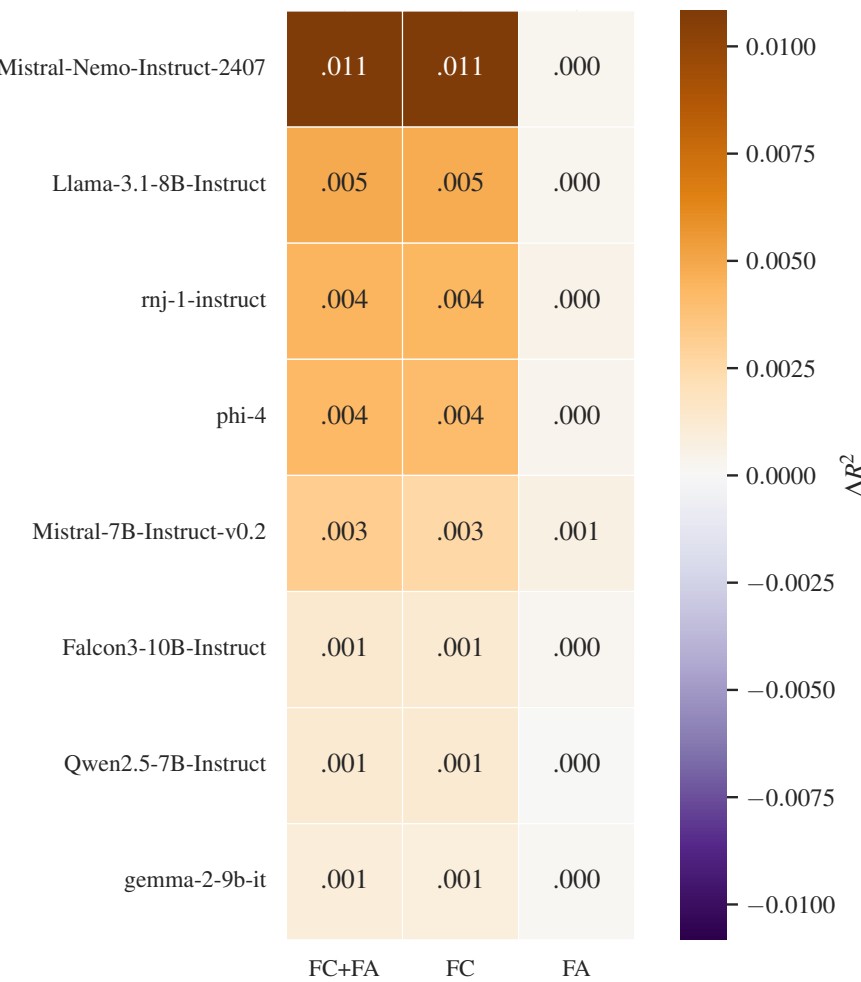

*Figure 15.* Ablation study: Incremental contribution of behavioral predictors to held-out-words ridge regression performance for non-mean-centered hidden states, reported as $\Delta R^2$ relative to a baseline including lexical and cross-model similarity features.

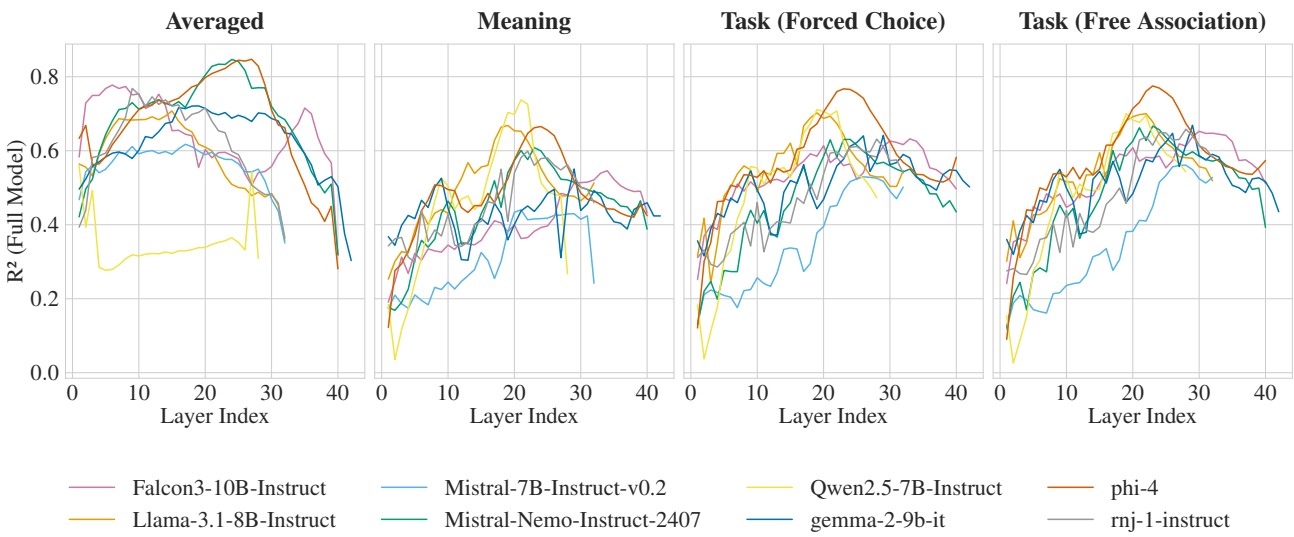

*Figure 16.* Ablation study: Layer-wise held-out-words ridge performance ($R^2$) for predicting each model's non-mean-centered hidden-state similarity from behavioral and lexical similarity features, shown separately for the four embedding extraction strategies.

