# OpenReview forum: "From Associations to Activations: Comparing Behavioral and Hidden-State Semantic Geometry in LLMs"
_ICML.cc/2026/Conference — ICML 2026 regular_

### Official Review · Reviewer_fvSh · 2026-03-01

**Soundness:** 4
**Presentation:** 3
**Significance:** 3
**Originality:** 3
**Overall Recommendation:** 4
**Confidence:** 3

**Summary:**

The authors use RSA and regression to compare the behavior of a model (probabilities of outputs over a set vocabulary) with the activation structures of the same model. They analyze two paradigms from psycholinguistics: free association and forced choice, and show that behavioral similarity can predict unseen hidden-state similarities.

**Compliance With Llm Reviewing Policy:**

Affirmed.

**Final Justification:**

I did not realize that this paper was submitted to the Neuroscience/Cognitive Science track. Given that, and the author's major contribution in the form of a detailed dataset of experimental psycholinguistic behavior in LLMs, I have raised my score; although I still believe that the paper is quite dense to read, and including more accessible discussion/interpretation in the main text (while still including full results in the Appendix) would make its contribution clearer to readers.

**Key Questions For Authors:**

1. What do you think is the main significance of your results to the field of machine learning? I understand the importance for cognitive science, but not yet for ML.
2. Is it surprising that cross-model consensus is a good predictor of a model's hidden-state similarity? (Similarity with what, other models, or baselines?)

**Limitations:**

Yes

**Strengths And Weaknesses:**

The question tackled in this paper is quite interesting: the relationship between model hidden state geometry and observed behavior. The paper is extremely thorough and detailed, with enough methodological details to replicate the results, eight models analyzed, four separate hidden-state extraction strategies, and ample baselines (the experimental design seems very sound to me). The setup is original to my knowledge, in terms of comparing psycholinguistic behavior to internal representations. The presentation is organized and well-written, albeit quite dense (related to criticism below).

My main concern has to do with significance. Although experiments are extremely detailed and thorough, I am not quite sure what we learn from any of them. It feels like there are many results, but not enough "digestion" of those results. This may actually be an issue with presentation: for example, in Section 4.1, it is very difficult to integrate and gain any insights from all of these results for so many experimental conditions. If I were writing this paper, I would pick out only a subset of results (leaving the rest in the Appendix), and dedicate more space to discussion of what these results teach us. To me, it seems like the main insight (that behavioral observation preserves a lot of information about a system's internal hidden state geometry) is more relevant to cognitive science than to ML.

Rather than extensive baselines/alternate conditions, I would have appreciated more investigation into interesting patterns suggested by your results, like how task/meaning prompts have stronger alignment than natural contexts. From an ML-interpretability perspective, this seems like something that could have been interesting to investigate in depth.

---

> ### Author Rebuttal · Authors · 2026-03-30
>
> Although we regret to hear that you are not convinced of the significance of our work, we thank you for your review!
>
> > What do you think is the main significance of your results to the field of machine learning? I understand the importance for cognitive science, but not yet for ML.
>
> Let us try to clarify the scope of our work.
>
> First of all, we would like to highlight that this paper was intentionally submitted to the **"Applications → Neuroscience, Cognitive Science"** track. The core question, inferring internal representations from observable behavior, is central to cognitive science, where internal states are not directly accessible (Kumar, 2021). FC and FA are the standard paradigms for estimating mental representations, and our setup provides the first large-scale evaluation of whether this inference holds in LLMs. Additionally, our dataset of 17.5 million trials across 8 LLMs *will be the largest openly released dataset of experimental psycholinguistic behavior in LLMs*. The Applications track explicitly highlights datasets as a form of contribution (cf. Call for Papers). Applications of psycholinguistic behavioral data in ML research include, among others, enriching ImageNet with similarity judgments from FC behavior (Roads & Love, 2021) and using LLM-generated FA data to study LLM biases (Abramski et al., 2024). Our goal was hence, first and foremost, to contribute to the *intersection of ML & cognitive science*, and we think that the ICML Call for Papers, which explicitly encourages "applications such as healthcare, physical sciences, biosciences, social sciences, sustainability, and climate etc." (cf. Call for Papers), justifies us in our submission.
>
> Apart from that, and despite not being the main focus, we see three concrete *ML takeaways*:
>
> (a) *Black-box interpretability*: FC recovers internal semantic geometry using only discrete text outputs (mean RSA r=.43 across all extraction strategies, layers, and models). This applies to any model behind an API without hidden-state or logit access.
>
> (b) *Task design as an interpretability variable*: The FC >> FA differential (3× RSA) is non-obvious: constrained comparison tasks yield far more faithful readouts than open-ended generation.
>
> (c) *Model-specific signal*: In a fingerprinting analysis with three-way train/test/feature separation, FC behavioral profiles carry substantially more model-specific signal than FA (0.99 vs. 0.59 accuracy, chance: 0.125).
>
> We regret not having made this more accessible in our manuscript, which we will revise for the camera-ready deadline.
>
> > Rather than extensive baselines/alternate conditions, I would have appreciated more investigation into interesting patterns suggested by your results, like how task/meaning prompts have stronger alignment than natural contexts.
>
> Regarding the differences in alignment between meaning-focused (Meaning, Task_FC, Task_FA) and natural embedding contexts (Averaged), we conducted additional quantitative analyses.
>
> Meaning-focused embeddings peak *earlier* in the network (≈30% depth), whereas averaged embeddings peak *later* (≈72%). In early and mid layers, meaning-focused FC alignment is 54% higher (.471 vs. .305). Crucially, layer-wise trajectories are tightly correlated across all measures (mean r = .85) under meaning-focused extraction, but decouple under averaged extraction (mean r = .37; see Figs. 7 and 9).
>
> We interpret this as meaning-focused prompts placing the model in a “word-in-focus” mode that encodes what a concept is at mid-depth, whereas averaged embeddings dilute this signal across 50 diverse contexts.
>
> We will expand this analysis into a dedicated subsection and move auxiliary results to the appendix in the camera-ready.
>
> > Is it surprising that cross-model consensus is a good predictor of a model's hidden-state similarity?
>
> No, and this is by design. Consensus is a deliberately strong upper-bound baseline: it aggregates continuous hidden-state information from seven other models across all layers. We assume that the high RSA alignment reflects the shared "platonic" geometry across LLMs (Huh et al., 2024). Nevertheless, FC, using only text outputs, adds predictive signal beyond consensus.
>
> We thank you for your review and hope we have addressed all your concerns and questions. Please let us know if further clarification is needed.
>
> ### References
> - Abramski, K., Lavorati, C., Rossetti, G., & Stella, M. (2024). LLM-generated word association norms. In HHAI 2024 (pp. 3-12). IOS Press.
> - Huh, M., Cheung, B., Wang, T., & Isola, P. (2024). The platonic representation hypothesis. arXiv preprint arXiv:2405.07987.
> - Kumar, A. A. (2021). Semantic memory: A review of methods, models, and current challenges. Psychonomic Bulletin & Review, 28(4), 1283-1317.
> - Roads, B. D., & Love, B. C. (2021). Enriching ImageNet with human similarity judgments and psychological embeddings. In Proceedings of the IEEE/CVF Conference on Computer Vision and Pattern Recognition (pp. 3547-3557).

---

> > ### Author Rebuttal · Reviewer_fvSh · 2026-04-02
> >
> > Thank you for your careful response to my questions. Given the fact that this was submitted to the applications in neuroscience and cognitive science track, and your clear contribution in the form of a new dataset, I now understand the contribution of your work. I still do think that not all of the analysis needs to be presented in the main text; readers would benefit from discussion that is a little bit more broad and ties your findings together. But I will raise my score.

---

> > > ### Author Response · Authors · 2026-04-04
> > >
> > > Thanks a lot for your response and for reconsidering your initial evaluation, we greatly appreciate it! We agree that the findings of our work could be organized in a more streamlined way, and will adjust the manuscript to make our contributions more accessible.

---

### Official Review · Reviewer_RdAZ · 2026-03-11

**Soundness:** 3
**Presentation:** 3
**Significance:** 4
**Originality:** 4
**Overall Recommendation:** 5
**Confidence:** 3

**Summary:**

The author proposes to rebuild the internal semantic geometric structure of LLMs through their behavioral outputs when the hidden states are inaccessible. Two paradigms FA and FC are introduced from cognitive psychology experimental methods, and results demonstrate that the FC mode achieves performance better than baselines including BERT and FastText on this task, with the uncovered semantic geometric structure aligning with the ground-truth geometry derived from internal model activations. Further, the paper suggests that all LLMs may share a homomorphic fundamental semantic geometric structure.

**Compliance With Llm Reviewing Policy:**

Affirmed.

**Final Justification:**

The paper is novel and sound, and the rebuttal is satisfactory to me. Therefore, I will keep the positive rating unchanged.

**Key Questions For Authors:**

- In practical applications, base models are frequently finetuned to specific domains. Would you expect such finetuning to primarily shift the model's behavioral geometry and its alignment to the hidden states?

- The transition from continuous hidden states to discrete text outputs inevitably introduces non-linear un-embedding and sampling truncation, leading to information loss. Have you considered computing the semantic geometry derived directly from the full logit distributions (over the candidate sets) to serve as a 'theoretical upper bound' and quantify the 'information loss' of the proposed method?

**Limitations:**

Yes

**Strengths And Weaknesses:**

# Strengths
- The paper focuses on an important and practical question: how to evaluate the hidden state geometry of closed-source LLMs (such as GPT, Claude, Gemini, etc.), and show hopeful results with the help of psychological paradigms, which can be valuable for the mechanism interpretation community.

- The experiments covers over 17.5 million experimental trials across eight instruction-tuned large language models. Also, multiple measurements such as RSA and nearest-neighbor overlap gives consistent findings.

- The paper is well written and easy to follow.

# Weaknesses
- The ridge regression part seems to contradict with the motivation. The regression tries to show the (moderate) extra information gain of behavioral reconstruction in addition to the baselines. However, the strength of behavioral reconstruction is its independence of hidden state accessibility. As a result, the overlap between behavioral and hidden states info is not important. I think this part should be re-organized to demonstrate the on-par regression power of the two.

- Figure 3 shows that the RSA and NN@k performances of FC falls behidn cross-model consensus by a large margin, which indicates that the proposed method still requires performance improvement.

---

> ### Author Rebuttal · Authors · 2026-03-30
>
> We thank the reviewer for the positive assessment and for recognizing the practical value of our approach for model interpretability.
>
> > The ridge regression part seems to contradict with the motivation. The regression tries to show the (moderate) extra information gain of behavioral reconstruction in addition to the baselines. However, the strength of behavioral reconstruction is its independence of hidden state accessibility.
>
> Thank you for raising this point. We believe direct RSA comparison demonstrates FC's standalone value. Mean RSA across all four extraction strategies, all layers, and all eight models is:
>
> - FC: r = .43
> - Cross-model consensus: r = .75
>
> FC recovers 57% of consensus's alignment using only discrete text outputs. For any realistic black-box scenario (e.g., probing Opus 4.6), the open-source models underlying consensus are always available -- one would naturally use them. The unique value of FC is not *replacing* consensus but *complementing* it with model-specific behavioral signal that consensus cannot capture by construction (see fingerprinting result in our response to reviewer fvSh). We will restructure Section 4.3 to lead with this direct RSA comparison.
>
> > Figure 3 shows that the RSA and NN@k performances of FC falls behind cross-model consensus by a large margin, which indicates that the proposed method still requires performance improvement.
>
> We agree that a gap exists, but its interpretation is important. Cross-model consensus is a strong internal baseline: it aggregates hidden-state similarities across multiple models and layers, and thus has access to rich representational information. In contrast, FC relies only on discrete behavioral outputs. The comparison therefore reflects a discrete black-box vs. continuous internal-representation setting, where a gap is expected. Importantly, we checked whether the gap can be attributed to noisy measurement. We verified that FC behavior is highly reliable: split-half correlations average r = .80 across models (Spearman–Brown corrected: r = .88), reflecting the large number of trials per cue. This indicates that the behavioral similarity estimates are stable. However, more sophisticated behavioral embedding methods such as joint kernel-metric learning (Roads & Mozer, 2019) could yield higher-fidelity geometries. Bridging this gap is a key direction connecting ML interpretability with cognitive measurement methodology.
>
> > In practical applications, base models are frequently finetuned to specific domains. Would you expect such finetuning to primarily shift the model's behavioral geometry and its alignment to the hidden states?
>
> Thanks for the question! We expect fine-tuning to shift both geometries, with alignment preserved: behavioral outputs are generated *from* hidden states via the unembedding layer, so reshaping one reshapes the other. This needs empirical validation, but Binz et al. (2024) provide supporting evidence: fine-tuning Llama-3.1-70B on purely behavioral data (~10M human choices) shifts internal representations toward better alignment with human fMRI activity — without any neural training signal. FC candidate sets can be adapted to domain-specific vocabularies, making this a natural next step.
>
> > Have you considered computing the semantic geometry derived directly from the full logit distributions (over the candidate sets) to serve as a 'theoretical upper bound' and quantify the 'information loss' of the proposed method?
>
> We agree that leveraging logits is an important direction. Using discrete outputs was a deliberate choice to reflect a setting in which logits are not available. Our FC data uses greedy decoding; computing logit distributions for all 17.5M trials would be substantially more expensive. The high split-half reliability (Spearman–Brown r = .88) shows that FC measurements are stable. Quantifying potential gains from logits is an interesting direction for future work.
>
> Thank you for your review. We hope to have addressed all your questions. Please let us know if further clarification is needed.
>
> ### References
> - Binz, M., Akata, E., Bethge, M., Brändle, F., Callaway, F., Coda-Forno, J., ... & Schulz, E. (2024). Centaur: a foundation model of human cognition. arXiv preprint arXiv:2410.20268.
> - Roads, B. D., & Mozer, M. C. (2019). Obtaining psychological embeddings through joint kernel and metric learning. Behavior Research Methods, 51(5), 2180-2193.

---

> > ### Author Rebuttal · Reviewer_RdAZ · 2026-04-02
> >
> > The authors' reply solves most of my concerns. I am still curious about the logit distributions results, but this does not undermine the overall quality of this paper.

---

> > > ### Author Response · Authors · 2026-04-02
> > >
> > > We thank the reviewer for their positive follow-up and for confirming that our responses have addressed their concerns. We also appreciate your continued interest in the logit-based analysis.
> > >
> > > We agree that directly comparing behavioral geometry to logit-derived similarity would provide a valuable complementary perspective. As noted, our current design focuses on the black-box setting where logits are not accessible, but extending the framework to incorporate logit distributions is a natural and promising direction for future work. We will highlight this more explicitly in the camera-ready version.
> > >
> > > Thank you again for your thoughtful engagement and for your supportive assessment of our work.

---

### Official Review · Reviewer_TCUJ · 2026-03-12

**Soundness:** 3
**Presentation:** 3
**Significance:** 1
**Originality:** 1
**Overall Recommendation:** 3
**Confidence:** 2

**Summary:**

This paper studies the relationship between behavior and “latent semantic space” of language models. To correlate the two, the authors design a “behavior similarity matrix” and a “hidden-state similarity matrix”. The behavior similarity matrix is constructed by providing a language model with a set of cues, and observing which candidate words are selected (associated) with the cue token. This in turn produces a sparse representation per cue, after which one can measure how similar the cue representations are with one another.

Similarly, for hidden-state similarity matrix, the model is given contexts containing cue words, from which some intermediate representation corresponding to the cue words are extracted. Similarly, one can measure how similar such hidden-state representations are amongst the cue words to construct a hidden-state similarity matrix.

Given these two objects, the authors then study the similarity of them using RSA (pearson correlation) or a k-nearest neighbor analysis.

**Compliance With Llm Reviewing Policy:**

Affirmed.

**Final Justification:**

Thank you for your response.

The additional context is indeed helpful, and helps me better understand the contributions. I regret to say that I am left a bit unsatisfied. The rebuttals have clarified some of the contributions that I have previously missed, including that the work provides practical guidance on how to assess the similarities in behavior vs. internal representations, as well as the large-scale trial dataset (based on the discussions with other reviewers). Regarding the first point (practical guidance), the paper's meticulous experimental designs support such claims. On the other hand, regarding the second point (release of large-scale data), I understand that such artifacts align with the CfP but I'm not sure I understand what the implications of this dataset is for the field. To be clear, I say this without having a background in cog-sci / neuroscience.

While these contributions have been made clearer, I am still struggling to come up with a concise summary of what new insights I have learned from reading the work. If it is the fact that behavior can be a proxy for internal representations, this isn't surprising to me (authors and I perhaps just have different priors). If it's the fact that FC works better than FA, maybe I just lack the context to appreciate the significance of this.

With that being said I am maintaining my score but am lowering my confidence, and I trust that the AC will take this into account.

**Key Questions For Authors:**

1. I’m a bit confused as to how the candidate words are chosen for the forced-choice paradigm, but also why it requires multiple FC trials per cue.
2. I’m a bit confused about the design of cross-model consensus, specifically why representations are averaged across layers, but also why it has such a high RSA score in Figure 3.

**Limitations:**

yes

**Strengths And Weaknesses:**

Strengths:
The experimental setups are very carefully constructed with a lot of details provided, and the text is well written with clear structure.

Weaknesses:
a) Perhaps because I come from a non-cognitive science background, the motivation of the paper (including some of the experimental setups) is a bit unclear. As I understand it, the motivation is to study the similarity between observed behaviors of LLMs versus its latent states. While an interesting idea, there’s a bit of a nuance in that the objects being studied are “behavior similarity embeddings” versus a “hidden state similarity embedding”, and I don’t understand why studying the correlation between these two objects is useful. Put differently, because the two spaces being correlated are the “behavior similarity space” and “hidden state similarity space”, not “behavior space” and “hidden-state space”, I’m not sure what to do with the findings - what are the newly learned insights or applications that I can draw?

Put differently, the paper currently reads like “this comparison is something do-able for anyone interested” but does not offer a clear takeaway based on their results. Namely, should the results be taken as high similarity or not? Currently the only conclusive claims being made are comparisons between the methods (e.g., the forced-choice setting leads to higher RSA than free-association), but as a reader it’s not clear what to make of these numbers.

b) I appreciate how the authors have conducted their study, but I think the results are not surprising?
Afterall, the training objective of LLMs is to align similar contexts that share the same set of next tokens close to one another, which I think is essentially what the authors are trying to measure?
One argument against this point might be that the authors are studying mid-layers, not the last layer (from which the context embeddings are mapped to vocabulary space). However, we have seen from numerous works that there is a close relationship (ie high similarity) between mid-layers and the last layer (e.g., “Logit Lens”, SVCCA [2], steering works [3] and others [4, 5]).

c) The paper is squarely an empirical paper, and the paper consists of multiple experiments but ultimately I think only two of them offer substance towards the authors’ narratives (Figure 3a, Figure 5), while the others feel like auxiliary studies that could end up in the appendix. Unfortunately, I don’t think this meets the expected contributions for ICML.

[1] https://www.lesswrong.com/posts/AcKRB8wDpdaN6v6ru/interpreting-gpt-the-logit-lens
[2] https://arxiv.org/pdf/1706.05806
[3] https://arxiv.org/pdf/2312.06681
[4] https://arxiv.org/pdf/2503.21073
[5] https://arxiv.org/pdf/2512.19941

---

> ### Author Rebuttal · Authors · 2026-03-30
>
> We thank the reviewer for their review and for noting the quality of the experimental design.
>
> > I don't understand why studying the correlation between these two objects is useful. [...] the two spaces being correlated are the "behavior similarity space" and "hidden state similarity space", not "behavior space" and "hidden-state space".
>
> We would like to clarify that comparing similarity structures is *the* methodologically appropriate approach when representations live in fundamentally different spaces (Kriegeskorte et al., 2008). Behavioral responses are sparse count vectors; hidden states are dense continuous embeddings. These differ in dimensionality, basis, and scale. RSA compares relational structure: which items are close or far, invariant to rotation or scaling. This is standard in cognitive science and ML (e.g., Kornblith et al., 2019, ICML).
>
> Conceptually, our goal is to test whether observable behavior preserves information about internal representations—a central problem in cognitive science, where internal states must be inferred from behavior (Kumar, 2021). Association tasks are the standard tools for this, and our setup enables a direct test in LLMs. We thus interpret the correlation between FC and hidden-state similarity as evidence for behavior as a proxy for internal representational geometry. This motivation underlies our submission to the **"Applications → Neuroscience/Cognitive Science"** track. Furthermore, our 17.5M-trial dataset will be, to the best of our knowledge, the *largest openly released dataset of experimental psycholinguistic behavior for LLMs*. We will expand on this point in the camera-ready.
>
> > I think the results are not surprising? Afterall, the training objective of LLMs is to align similar contexts that share the same set of next tokens close to one another.
>
> We believe this interpretation overlooks that, if alignment were trivially implied by the next-token training objective, FC and FA should recover geometry similarly, since both probe next-token behavior. However, we observe a ~3× difference in RSA. The key reason is that FC *constrains the response space* to a shared candidate set, forcing explicit similarity judgments and concentrating probability mass. Thus, the FC > FA result reflects a measurement effect: only tasks that constrain responses expose the underlying similarity geometry.
>
> > I think only two of them offer substance towards the authors' narratives (Fig. 3a, Fig. 5), while the others feel like auxiliary studies that could end up in the appendix. Unfortunately, I don't think this meets the expected contributions for ICML.
>
> We respectfully disagree. Each analysis answers a question the others cannot. RSA tests global alignment; NN@k tests local neighborhoods. Regression tests generalization to unseen words. The extraction strategy breakdown (Fig. 3b) shows that alignment varies with prompting context and peaks at different layers, revealing when behavior reflects underlying hidden states. Per-model results (Fig. 4) establish generality across all 8 models. Removing any one leaves an evidential gap.
>
> > I'm a bit confused as to how the candidate words are chosen for the forced-choice paradigm, but also why it requires multiple FC trials per cue.
>
> Thank you for pointing this out. For each cue, the remaining 4,999 items are partitioned into non-overlapping groups of 16, yielding 313 trials per cue. Each trial selects 2 winners from 16 candidates, producing a binary signal. Multiple trials are necessary to build up sufficient density in the behavioral matrix for computing meaningful pairwise cosine similarities. We will make this more explicit in the camera-ready.
>
> > I'm a bit confused about the design of cross-model consensus, specifically why representations are averaged across layers, but also why it has such a high RSA score in Figure 3.
>
> We would like to note that consensus for model *m* averages hidden-state cosine similarity across all layers and all *other* models, excluding *m*. This averaged signal should reflect the shared "platonic" geometry across LLMs (Huh et al., 2024). The high RSA is expected by design as a strong upper-bound baseline for continuous hidden-state structure. FC adds signal beyond this shared structure.
>
> We appreciate your review and hope to have addressed your concerns! Please reach out if anything needs further clarification.
>
> ### References
> - Huh, M., Cheung, B., Wang, T., & Isola, P. (2024). The platonic representation hypothesis. arXiv preprint arXiv:2405.07987.
> - Kornblith, S., Norouzi, M., Lee, H., & Hinton, G. (2019). Similarity of neural network representations revisited. In International Conference on Machine Learning (pp. 3519-3529). PMLR.
> - Kriegeskorte, N., Mur, M., & Bandettini, P. A. (2008). Representational similarity analysis. Frontiers in Systems Neuroscience, 2, 4.
> - Kumar, A. A. (2021). Semantic memory: A review of methods, models, and current challenges. Psychonomic Bulletin & Review, 28(4), 1283-1317.

---

> > ### Author Rebuttal · Reviewer_TCUJ · 2026-03-31
> >
> > Thank you for your response!
> >
> > > We would like to clarify that comparing similarity structures is the methodologically appropriate approach when representations live in fundamentally different spaces
> >
> > Point well taken that your approach allows to compare structures across fundamentally different spaces, and that similar methods are often used in our fields (CKA, kernels, etc.).
> >
> > Perhaps what I wish to understand is what to take away from the study. I do not wish to sound dismissive, but the draft currently reads as "we tried a set of representation comparison methods and some experimental conditions led to higher alignment than others". I understand that the question is whether behavior can be a proxy for internal structure, so is the takeaway you wish to provide "yes, but only under specific settings"?
> >
> > It seems like this sentiment is shared by reviewer fvSh: "Although experiments are extremely detailed and thorough, I am not quite sure what we learn from any of them. It feels like there are many results, but not enough "digestion" of those results."
> >
> > > Thus, the FC > FA result reflects a measurement effect: only tasks that constrain responses expose the underlying similarity geometry.
> >
> > Perhaps I am coming with different priors, but I still do not think the results are surprising in that given similar behavior, one can find similar underlying representational structures. Your response that FC > FA reveals that only tasks with constrained responses expose the underlying similarities is understood, but I don't think it explains why the results would have been unexpected. Perhaps the contribution you're trying to provide is a practitioner's guide on how to do these types of studies?
> >
> > While I cited a few papers in my original response that shaped my priors, I think the platonic representation hypothesis work that you cite is also indicative that we would expect similar representations under similar behaviors. If we just have different priors, we can agree to disagree and move on from this point.
> >
> > RE: cross-model consensus:
> >
> > I think I understand now, thank you for the clarification. I think these are additional papers that you could cite to motivate the cross-model consensus setup: https://arxiv.org/abs/2504.08775, https://arxiv.org/abs/2503.21073

---

> > > ### Author Response · Authors · 2026-04-01
> > >
> > > Thank for the follow-up, we appreciate the way you have articulated your perspective, which helped us clarify the contribution. We will cite both suggested papers (arXiv:2504.08775, arXiv:2503.21073) in the camera-ready.
> > >
> > > **On the takeaway:**
> > >
> > > > Perhaps what I wish to understand is what to take away from the study. I do not wish to sound dismissive, but the draft currently reads as "we tried a set of representation comparison methods and some experimental conditions led to higher alignment than others". I understand that the question is whether behavior can be a proxy for internal structure, so is the takeaway you wish to provide "yes, but only under specific settings"?
> > >
> > > We understand your concern, and practical guidance is certainly part of what we offer. However, we believe the contribution goes beyond this. The key insight is methodological: task design fundamentally shapes what can be recovered about internal representations.
> > >
> > > To put this in context: FA and FC (including variants like odd-one-out and triadic comparison) are the two most widely used paradigms for studying mental representations in cognitive science. FA has a 145+ year history (Galton, 1879), and the Small World of Words project alone has collected 15M+ responses across 14 languages (De Deyne et al., 2019). On the FC side, the THINGS initiative collected 4.7M triplet judgments from 14,000+ participants (Hebart et al., 2023). Enormous research effort has gone into both paradigms for mapping semantic knowledge.
> > >
> > > Even though FA, as the classical paradigm, has long attracted more attention than FC paradigms, our findings show that FC substantially outperforms FA for recovering internal geometry. FA captures associative network structure (which words activate which), but its unconstrained response space produces sparse, heavy-tailed distributions that lose geometric fidelity. FC forces explicit comparisons, reducing sparsity and yielding similarity estimates that more faithfully reflect the underlying geometry. We show this across global alignment (RSA), local neighborhoods (NN@k), and generalization to new words (regression).
> > >
> > > This has broader implications for cognitive science. Reducing the response space to a shared candidate set provides better read-outs of what can be learned about the system, whether that system is a human mind or a language model.
> > >
> > > We hear the "digestion point" (shared by reviewer fvSh) and will restructure the camera-ready to lead with these insights rather than presenting results condition-by-condition. That should make the paper much easier to read.
> > >
> > > **On priors:**
> > >
> > > > While I cited a few papers in my original response that shaped my priors, I think the platonic representation hypothesis work that you cite is also indicative that we would expect similar representations under similar behaviors. If we just have different priors, we can agree to disagree and move on from this point. [...] Perhaps I am coming with different priors, but I still do not think the results are surprising in that given similar behavior, one can find similar underlying representational structures.
> > >
> > > We agree that the platonic representation hypothesis suggests we should expect similar representations under similar behaviors, and we understand why this may make the result seem unsurprising given your priors. Whether the FC > FA difference is surprising is ultimately a matter of perspective, and we are happy to move on from that point. We would like to emphasize that even when a result aligns with expectations, it still requires empirical validation, especially when it concerns a central methodological question in the field (here: cognitive science). That is what we provide here.
> > >
> > > Thank you again for your careful feedback! We hope this clarification better conveys the intended contribution and addresses your concerns.
> > >
> > > ### References
> > > - De Deyne, S., Navarro, D. J., Perfors, A., Brysbaert, M., & Storms, G. (2019). The “Small World of Words” English word association norms for over 12,000 cue words. Behavior research methods, 51(3), 987-1006.
> > > - Galton, F. (1879). Psychometric experiments. Brain, 2(2), 149-162.
> > > - Hebart, M. N., Contier, O., Teichmann, L., Rockter, A. H., Zheng, C. Y., Kidder, A., ... & Baker, C. I. (2023). THINGS-data, a multimodal collection of large-scale datasets for investigating object representations in human brain and behavior. eLife, 12, e82580.

---

### Official Review · Reviewer_uiCy · 2026-03-13

**Soundness:** 3
**Presentation:** 3
**Significance:** 2
**Originality:** 3
**Overall Recommendation:** 4
**Confidence:** 3

**Summary:**

This paper investigates whether the semantic geometry of LLM hidden states can be recovered from the model’s overt behavior in psycholinguistic-style tasks. The authors construct large-scale behavioral spaces from two association paradigms, forced choice and free association, and compare them with layer-wise hidden-state similarity structures across multiple language models. The main finding is that behavioral data does contain recoverable information about internal semantic organization, but the strength of this correspondence depends strongly on task design. In particular, forced-choice behavior aligns much better with hidden-state geometry than free association. The paper is interesting because it connects cognitive science ideas about inferring latent representations from behavior with modern LLM interpretability.

**Compliance With Llm Reviewing Policy:**

Affirmed.

**Final Justification:**

Thank you for the detailed rebuttal. I appreciate the authors’ clarifications, and I find the response to my concerns helpful and thoughtful. The rebuttal has improved my overall assessment of the paper.

That said, I would still encourage the authors to clarify one point more explicitly in the paper: if the dataset is intended to be presented as a core contribution, its value should be articulated in terms of how it can be reused beyond serving this particular study. In other words, it would strengthen the paper to explain more clearly where this dataset may be useful as a community resource, evaluation set, or benchmark, rather than emphasizing the number of trials alone.

Overall, I appreciate the authors’ response and will raise my score accordingly.

**Key Questions For Authors:**

1. The current analyses show that behavioral measurements can recover part of the hidden-state geometry, but could the authors further clarify what kind of semantic structure is actually being captured? For example, can they disentangle whether the recovered geometry mainly reflects conceptual similarity, lexical association statistics, prompt-induced formatting effects, or some combination of these factors?

2. Since cross-model consensus already explains a large fraction of the hidden-state geometry, could the authors better characterize what unique information is contributed by behavior beyond this shared structure? For instance, are there specific semantic domains, word categories, or layers where behavioral data provides clearer gains that are not already captured by other models?

3. The current study focuses on word-level association tasks. Do the authors expect the main conclusions to hold for richer semantic units such as phrases or sentences, or for more naturalistic behavioral tasks? It would be helpful if the paper could discuss this limitation more explicitly, or provide preliminary evidence on whether the same behavior–representation relationship extends beyond isolated words.

**Limitations:**

Yes

**Strengths And Weaknesses:**

**Strengths:**

1. The paper studies a genuinely interesting question at the intersection of cognitive science and LLM interpretability: whether external behavior can serve as a proxy for internal semantic structure.

2. One practically useful takeaway is that task design matters a lot. The finding that forced-choice judgments provide a more stable readout of internal geometry than freer generation-based responses is valuable for future work on black-box model analysis.

**Weaknesses:**

1. Much of the evidence is based on representational similarity and predictive alignment, but the paper is less clear about what specific aspects of semantics are being captured. It remains somewhat ambiguous whether the recovered structure reflects conceptual knowledge, lexical associations, prompt artifacts, or a mixture of these factors.

2. The very strong performance of cross-model consensus suggests that a large fraction of the explainable geometry may come from generic shared structure across models, which makes the incremental contribution of behavior comparatively modest. This slightly weakens the claim that behavior provides a uniquely informative window into hidden states.

3. The study is limited to word-level association settings. It is not yet clear whether the main conclusions would generalize to richer semantic units such as phrases, sentences, or more naturalistic tasks.

---

> ### Author Rebuttal · Authors · 2026-03-30
>
> We thank the reviewer for their review and for recognizing the originality, soundness, and presentation of our work.
>
> > Could the authors further clarify what kind of semantic structure is actually being captured? For example, can they disentangle whether the recovered geometry mainly reflects conceptual similarity, lexical association statistics, prompt-induced formatting effects, or some combination?
>
> Let us address this question from two perspectives.
>
> *Functional operationalization*: Our approach adopts a *purely functional* stance: we measure the representational structure that directly determines model output, i.e., its behavior. This represents a mixture of lexical association statistics and conceptual similarity. This functional operationalization is the standard approach in cognitive science, where FC similarity judgments and FA are the primary paradigms for measuring semantic memory representations (Kumar, 2021).
>
> *Prompt-induced artifacts*: Being aware of the problem of prompt artifacts, we decided to compare four embedding extraction strategies. We see that the trajectories of the meaning-focused embeddings (meaning, Task_FC and Task_FA) are very similar across *all* measures (mean r=.85), while we see a decoupling of the trajectories under averaged extraction (mean r=.37). This suggests that the meaning-focused embeddings are capturing a more stable "word-in-focus" semantic structure, while the averaged embeddings are capturing a more diffuse structure reflecting the averaging of 50 sampled contexts.
>
> > Could the authors better characterize what unique information is contributed by behavior beyond this shared structure? For instance, are there specific semantic domains, word categories, or layers where behavioral data provides clearer gains?
>
> We ran a stratified held-out-word prediction analysis (baseline: FastText + BERT + consensus; full: baseline + FC + FA) to isolate behavioral gains beyond shared structure.
>
> *Word categories (ΔR²)*: Gains vary systematically across WordNet categories. *Concrete and experiential* categories show the largest gains: food (+.10), feeling (+.04), animal (+.03), person (+.03), plant (+.03). Abstract, functionally defined categories show the smallest: communication (+.01), cognition (+.01), artifact (+.01). This suggests that behavioral signal is *most complementary* to baselines for concrete/experiential categories.
>
> *Frequency (ΔR²)*: *Low-frequency* words benefit most from behavioral signal (+.02), while mid-frequency (+.01) and high-frequency (+.01) words benefit less. This suggests behavior is especially informative where distributional statistics from pretraining are sparser.
>
> *Layers (ΔR²)*: There is no overall linear trend across depth (r = .03, p = .40), but prompt-specific patterns diverge: for meaning-focused extraction (template, FC, FA prompts), behavioral gains are 2-3× larger in early/mid layers than late layers (e.g., template: early +.007 vs late +.002). For averaged extraction, gains increase with depth (+.04 to +.06). This mirrors the dissociation in point 1.
>
> We will present the full stratified analysis in the camera-ready version.
>
> > Do the authors expect the main conclusions to hold for richer semantic units such as phrases or sentences, or for more naturalistic behavioral tasks?
>
> We expect the qualitative pattern to generalize. Our FC > FA explanation is *measurement-theoretic*: FC constrains responses to fixed candidate sets, reducing sparsity. Prior work has collected phrase- and sentence-level similarity judgments (Gershman & Tenenbaum, 2015; Arana et al., 2023), confirming behavioral paradigms extend to richer levels. However, those representations are more compositional, so the magnitude of alignment may differ and needs empirical validation.
>
> Finally, we wanted to stress that this work was submitted to the **Applications → Neuroscience/Cognitive Science** track, connecting cognitive science ideas with LLM interpretability, as the reviewer recognizes. Additionally, our dataset of 17.5M trials will, to the best of our knowledge, be the *largest openly released dataset of psycholinguistic behavior of LLMs*. Datasets as contributions were explicitly encouraged by the Applications track (cf. Call for Papers). We hope this clarifies our submission rationale.
>
> Thank you again for taking the time to review our work! We hope to have addressed your concerns. Please let us know if further clarification is needed.
>
> ### References
> - Kumar, A. A. (2021). Semantic memory: A review of methods, models, and current challenges. Psychonomic Bulletin & Review, 28(4), 1283-1317.
> - Gershman, S. J., & Tenenbaum, J. B. (2015). Phrase similarity in humans and machines. In Proceedings of the Annual Meeting of the Cognitive Science Society (Vol. 37).
> - Arana, S., Hagoort, P., Schoffelen, J. M., & Rabovsky, M. (2024). Perceived similarity as a window into representations of integrated sentence meaning. Behavior Research Methods, 56(3), 2675-2691.

---

> > ### Author Rebuttal · Reviewer_uiCy · 2026-04-03
> >
> > Thank you for the detailed rebuttal. I appreciate the authors’ clarifications, and I find the response to my concerns helpful and thoughtful. The rebuttal has improved my overall assessment of the paper.
> >
> > That said, I would still encourage the authors to clarify one point more explicitly in the paper: if the dataset is intended to be presented as a core contribution, its value should be articulated in terms of how it can be reused beyond serving this particular study. In other words, it would strengthen the paper to explain more clearly where this dataset may be useful as a community resource, evaluation set, or benchmark, rather than emphasizing the number of trials alone.
> >
> > Overall, I appreciate the authors’ response and will raise my score accordingly.

---

> > > ### Author Response · Authors · 2026-04-04
> > >
> > > Thank you for the response, we greatly appreciate that you reconsidered your initial evaluation! We agree that our manuscript will benefit from a more clear presentation of use-cases of our dataset. For instance, applications of psycholinguistic behavioral data in ML research include, among others, enriching ImageNet with similarity judgments from FC behavior (Roads & Love, 2021) and using LLM-generated FA data to study LLM biases (Abramski et al., 2024). We will revise the manuscript accordingly. Thanks!

---

### Review · Ethics_Reviewer_9cMb · 2026-04-08

**Recommendation:** No remediation action needed

**Ethics Issue:**

I didn't see an issue

---

### Decision · Program_Chairs · 2026-04-30

**Decision:**

Accept (regular)

**Comment:**

Inspired by studies in cognitive science, the paper studies whether we can infer something about the hidden-representation geometry of LLMs from their external behavior in pscyholinguistic tasks. The main finding is that this is possible if the task is designed to be forced choice (limits the LLM to choose from a restricted set of responses) rather than free-response. The paper also contributes a large-scale dataset of LLM behaviors.

The reviewers appreciated the contributions as valuable, specifically for black-box analysis of closed models (`uiCy`, `RdAz`)
and as an interesting (`uiCy`, `fvSh`) question that lies in the intersection of interpretability, LLMs and cognitive science. The reviewers also appreciated the care and the thoroughness of the empirical setup (`TCUj`, `fvSh`).

A number of concerns were recurring, and I hope that the authors find the feedback useful in improving the paper.

1. Much discussion has been around the significance and the takeaways of the paper.  `TCUj` found the motivation unclear (admitting their non-cognitive-science background), and `fvSh` opines that the results are not presented in a way that it's clear what to learn from them. Both `uiCy` and  `TCUj` get the impression that it is less surprising to find similarities between the representations of models in light of what we know about them (e.g., their shared objectives, platonic representation hypothesis etc.,) I found the authors response to this helpful (i.e., the findings here help, as noted by other reviewers, with blackbox interpretability and looking for model-specific signals.). Hopefully, the feedback can help the authors re-work the paper in a way that appeals to a wider (non-cogsci) audience. I also encourage the authors to think more deeply about `fvSh`'s feedback on the paper's presentation, which I believe is important and can improve the paper significantly. (Indeed, part of the contribution in a scientific manuscript lies in laying the emphasis at the right places and organizing results in a digestible way, rather than throwing a lot of results and hoping the reader can pick what sticks with them.)

2. Likewise, as `uiCy` concretely and thoughtfully points out, the paper would benefit from envisioning the value of the dataset beyond the specific questions studied in the paper. The authors have responded to this, which I hope will be included in future versions of the paper.

Overall, I recommend accepting the paper for its thoroughness and the practical guidance its provides for interpretability research.